



# Multi-year analysis of distributed glacier mass balance modelling and equilibrium line altitude on King George Island, Antarctic Peninsula

Ulrike Falk[1,2], Damián A. López[2,3], and Adrián Silva-Busso[4,5]

[1]Climate Lab, Institute for Geography, Bremen University, Germany
[2]Center for Remote Sensing of Land Surfaces (ZFL), Bonn University, Germany
[3]Institute of Geology and Mineralogy, University Cologne, Germany
[4]Faculty of Exact and Natural Sciences, University Buenos Aires, Argentina
[5]University of Buenos Aires (UBA), Buenos Aires, Argentina

*Correspondence to:* Ulrike Falk (ulrike.falk@gmail.com)

**Abstract.** The South Shetland Islands are located at the northern tip of the Antarctic Peninsula (AP). This region was subject to strong warming trends in the atmospheric surface layer. Surface air temperature increased about $3\,\mathrm{K}$ in 50 years, concurrent with retreating glacier fronts, an increase in melt areas, ice surface lowering and rapid break-up and disintegration of ice shelves. The positive trend in surface air temperature has currently come to a halt. Observed surface air temperature lapse rates show a high variability during winter months (standard deviations up to $\pm 1.0\,\mathrm{K}/100\,\mathrm{m}$), and a distinct spatial heterogeneity reflecting the impact of synoptic weather patterns. The increased mesocyclonic activity during the winter time over the past decades in the study area results in intensified advection of warm, moist air with high temperatures and rain, and leads to melt conditions on the ice cap, fixating surface air temperatures to the melting point. Its impact on winter accumulation results in the observed negative mass balance estimates. Six years of continuous glaciological measurements on mass balance stake transects as well as five years of climatological data time series are presented and a spatially distributed glacier energy balance melt model adapted and run based on these multi-year data sets. The glaciological surface mass balance model is generally in good agreement with observations, except for atmospheric conditions promoting snow drift by high wind speeds, turbulence-driven snow deposition and snow layer erosion by rain. No drift can be seen over the course of the 5-year model run period. The winter accumulation does not suffice to compensate for the high variability in summer ablation. The results are analyzed to assess changes in melt water input to the coastal waters, specific glacier mass balance and the equilibrium line altitude. The Fourcade Glacier catchment drains into Potter cove, has an area of $23.6\,\mathrm{km}^2$ and is to 93.8% glacierized. Annual discharge from Fourcade Glacier into Potter Cove is estimated to $\bar{q} = 25 \pm 6\,\mathrm{hm}^3/\mathrm{yr}$ with the standard deviation of 8 % annotating the high interannual variability. The average equilibrium line altitude (ELA) calculated from own glaciological observations on Fourcad Glacier over the time period 2010 to 2015 amounts to $ELA = 260 \pm 20$ m. Published studies suggest rather stable conditions of slightly negative glacier mass balance until the mid 80's with an ELA of approx. $150\,\mathrm{m}$. The calculated accumulation area ratio suggests dramatic changes in the future extent of the inland ice cap for the South Shetland Islands.




## 1 Introduction

Antarctic peripheral glaciers and ice caps cover an area of $132,867 \pm 6,643\,\text{km}^2$ and therefore represent a large fraction of all of earth's mountain glaciers and ice caps (Pfeffer et al., 2014). Changes in polar ice mass balance are observed as a direct consequence of changing atmospheric and oceanic conditions acting at different spatial and temporal scales, but also changes in internal dynamics (Payne et al., 2004; Davis et al., 2005; Van den Broeke et al., 2006). The Antarctic Peninsula (AP) has been warming with a rate exceeding $0.5\,\text{K}/decade$ at least during the last five decades (Skvarca et al., 1999; Vaughan et al., 2003; Steig et al., 2009; Barrand et al., 2013; Falk et al., 2016). This is likely caused by atmospheric and oceanic changes associated with the stratospheric ozone depletion (Fogt and Zbacnik, 2014), strengthening of the westerlies (Thompson and Solomon, 2002; Orr et al., 2008) and the increased presence of modified warm circumpolar deep water on the continental shelf (Joughin et al., 2014). This leads to changes in sea-ice season length, extent and concentration in the neighbouring seas (Parkinson, 2002; Abram et al., 2010; Stammerjohn et al., 2012). More specifically, the increase in near surface air temperature shows significantly higher warming trends during winter months for the South Shetlands (Falk et al., 2016). The east coast of the AP is mostly influenced by cold and dry air masses stemming from the adjacent Weddell Sea. By the contrary, the west coast jointly with the South Shetland Islands are directly exposed to the humid and relatively warm air masses from the South Pacific Ocean carried by the strong and persistent westerly winds. The interaction of the westerlies with the topography of the AP can lead to extreme Foehn events on the eastern side (Cape et al., 2015). Trends in surface air temperature in the AP have been analyzed and discussed in several studies obtained from meteorological observations of either manned or automatic weather stations (Doran et al., 2002; Turner, 2004; Turner et al., 2005; Chapman and Walsh, 2007; Barrand et al., 2013; Falk and Sala, 2015), sometimes complemented by satellite remote sensing data (Shuman and Stearns, 2001; Fahnestock et al., 2002; Steig et al., 2009). These studies congruently show the statistical significance of the observed positive trends in near-surface air temperatures of approx. $2.5°\text{K}$ over the last five decades along the AP, and the strong climatic change in the AP on a regional scale. Different drivers of the observed changes have been identified for winter and summer seasons: Winter sea ice concentration and mean sea level pressure anomalies are strongly connected with tropical variability, i.e. the El Niño Southern Oscillation (Bromwich et al., 2000; Yuan, 2004; Meredith et al., 2004), whereas changes of summer month's atmospheric circulation are driven by stratospheric ozone depletion and greenhouse gas concentrations (Thompson and Solomon, 2002; Perlwitz et al., 2008; Turner et al., 2009; Thompson et al., 2011). The seasonal variability is represented by the Southern Annular Mode (SAM) index and shows a variability that is of the same order of magnitude as the linear trend observed over the past 4 decades (Falk et al., 2016). As a consequence of the observed warming, striking glaciological changes have happened along the whole length of its western and eastern coasts. Studies along the AP show basal and surficial enhanced melting on ice shelves accompanied by subsequent collapse (Skvarca et al., 2004; Braun et al., 2009), widespread glacier acceleration and thinning (De Angelis and Skvarca, 2003), grounding line and calving front retreat (Rignot et al., 2011; Rau et al., 2004) among others. Ice shelves and glaciers of the Antarctic Peninsula region have been under a generalized retreat and disintegration trend at least during the last five decades (Rott et al., 1996; Skvarca et al., 1998; Scambos et al., 2000; Shepherd et al., 2003; Skvarca et al., 2004; Scambos et al., 2008; Braun et al., 2009). Ongoing atmospheric and cryospheric change are directly linked to



profound changes of the adjacent ocean (Meredith and King, 2005). Marine species in the Antarctic Peninsula region show extreme sensitivities to environmental conditions and their changes (Smith et al., 1999; Peck et al., 2004; Ducklow et al., 2007; Clarke et al., 2007; Montes-Hugo et al., 2009; Schloss et al., 2012; Quartino et al., 2013; Abele et al., 2017). Glacial melt water input to the coastal systems significantly changes physical and chemical properties, e.g. salinity, turbidity, light transmission and trace metals (Henkel et al., 2013; Sherrell et al., 2015). These studies rely on an accurate estimation of glacier melt. Systematic glaciological field studies are very scarce on both sides of the AP and in especially there are none that try to sufficiently capture inter- and intraannual variability. In this paper, we present a 6-year record of continuous glaciological observations obtained on very high temporal resolution to resolve winter melt periods and properly define the start of glacial accumulation and ablation periods. The time series is analysed with regard to climatic drivers and glacial melt is estimated to provide the necessary boundary conditions for interdisciplinary studies on the ongoing changes in the biota and species composition of the coastal waters in the South Shetland region. The following sections will

1. describe the meteorological observations, post-processing and gap filling,

2. define the hydrological catchment of the Warszawa Icefield draining into Potter cove,

3. assess calibration and validation of the glaciological model,

4. and the observed and specific mass balance and simulated glacial discharge to seasonal and interannual variability,

5. and finally, results of equilibrium line altitude and accumulation area ratio are used to assess future glacier extent.

## 2   Study area

King George Island (KGI) is the largest of the South Shetland Islands, located at 130 km from the northwestern tip of the AP. The coast and the slopes facing north are relatively uniform and smooth, whereas the south-facing coast and slopes are rugged and steep. Around 90 % of its 1250 km$^2$ are covered by a polythermal ice cap and are influenced by its maritime climate. Rückamp and Blindow (2012) have surveyed a significant part of it, finding that the mean ice thickness is approx. 240 m, with a maximum value of 422 m. The maximum elevation is 720 m a.s.l. in the central ice dome, with frequent secondary maxima of about 500-600 m a.s.l. across the island (Rückamp et al., 2011). The ice cap is divided into drainage basins according to the underlying geological structure (Braun and Hock, 2004). The different draining glaciers end either on land or as tidewater glaciers. Most of the glacier systems on the South Shetland Islands have shown significant retreats in recent past (Birkenmajer, 2002; Braun and Gossmann, 2002; Cook et al., 2005; Rückamp et al., 2011). The Warszawa Icefield covers the southwestern part of KGI including the Potter Peninsula. It includes two tidewater glaciers: Polar Club Glacier and Fourcade Glacier. The latter is draining into Potter Cove. Our data was mainly obtained on these two glaciers or in the neighboring areas (see map in Fig. 1). Jiahong et al. (1998) and Ferron et al. (2004) report an annual average surface air temperature of $-2.4°C$ and $-2.8°C$, respectively. Falk et al. (2016) show a strong positive trend in surface air temperature in especially in the winter, whereas for the summer month December the trend is slightly negative over the past four decades. In general, days with temperatures above freezing are rarely absent in winter and are frequent in summer. The occurrence of temperatures above the melting point all



year round is well recorded by thick and multiple ice lenses even in high elevations of the ice cap. Air temperatures above the melting point are generally associated with presence of low pressure systems and advection of warm and moist air from the mid latitudes over the surrounding oceans often resulting in precipitation in form of rain (Parish and Bromwich, 2007).

## 3    Data and glaciological model configuration

In this study, a physically based fully distributed energy balance model is applied to estimate glacier melt and glacier runoff into the Potter cove. The temporal resolution is defined by the resolution of the meteorological input data, which was sampled to hourly values. The spatial resolution is defined by the spatial input grids of the digital terrain model, the surface facies, aspect and slope etc. Both, a resolution of 50 m by 50 m using the digital terrain model (DTM) for KGI by Rückamp et al. (2011) and a resolution of 10 m by 10 m using the DTM by Braun et al. (2016) were applied. The output of these, as well runs using 100 m and 250 m resolution for the whole of King George Island, showed no significant differences.

### 3.1    Meteorological data sets

#### 3.1.1    Own meteorological measurements on Fourcade and Polar Club glacier, Potter Peninsula, KGI

An Automatic Weather Station (AWS) was installed in November 2010 at $S62°14'09.8''$ and $W58°36'48.7''$ at 196 m a.s.l., close to the approximate divide of Fourcade and Polar Club glaciers, which are both part of Warszawa Icefield. The AWS is equipped wind anemometers and vanes (Alpine Wind Monitor), air temperature and relative humidity sensors (HMP155A) at 1.4 and 2.5 m above ground, snow and ice temperature measurements (107 Thermistor Probes) installed at 10, 5 and 1 m depth in the glacier, and 0.1 and 0.3 m above ground to measure snow temperature during winter. The AWS included a four-component radiation sensor (NR01) for up- and downwelling long- and shortwave radiation fluxes, two narrow field infrared temperature sensors (IR120) facing Northwest and Southeast at a zenith angle of $40°$ to measure surface temperatures, and a sonic ranging sensor (SR50A) installed at an initial height of 1.9 m to measure surface elevation changes. For data acquisition and storage, a CR3000 Micrologger with extended temperature testing was used. The meteorological sensors were installed on a 3 m tripod that was fixed to 3 m aluminum poles drilled into the ice. The AWS is shown in Fig. 2 and 3. To ensure good quality of radiation measurements, all radiation sensors were mounted at a 3 m boom extended from the tripod and fixed to additional poles drilled into the ice. Leveling and adjustment of sensors were carried out according to ablation and accumulation. In case it was necessary outside of periods of summer field campaigns, this work was carried out by the overwintering Argentinean scientist. In particular at the end of the ablation season, the whole system needed to be lowered with a maximum of 2 to 3 m each year due to ablation at the AWS station. Power supply was realized with solar panels and a battery stack. Measurement rate was set to every 5 seconds with an averaging interval of 10 minutes. During the summer field campaign January - March 2012, an additional AWS (denoted as ZAWS) was installed in the accumulation area of the Warszawa Icefield at $S62°12'5.7''$ and $W58°34'58.4''$ at 424 m a.s.l. measuring wind speed and direction, air temperature and relative humidity, as well as downward shortwave radiation (Li190SB) for the time period of two weeks. All sensors and station equipment



were purchased from Campbell Scientific, Logan, Utah. Four additional air temperature and data logger sensors (UTL, Geotest Schweiz) were distributed on the investigated glaciers ($S62°14'32''$, $W58°35'54''$, 144 m a.s.l.; $S62°13'51''$, $W58°38'05''$, 65 m a.s.l.; $S62°13'58''$, $W58°38'28''$, 36 m a.s.l.) to assess the spatial variability and lapse rates of surface air temperature. One UTL sensor was kept at 2 m height at the AWS site to ensure the continuity of air temperature records during power failure

of the AWS. The meteorological data time series and climatology of the Potter Peninsula are discussed in detail by Falk et al. (2016). Orheim and Govorukha (1982) found layers of pyroclastic material in ice cores from the King George Island ice cap, dating them to vulcanic eruptions on Deception Island in December 1967, February 1969 and August 1970. These layers are surfacing in the ablation zone of the westward facing side of the Fourcade glacier on Potter Peninsula during melt periods. The dark material significantly changes the surface albedo (see Fig. 3) The resulting impact on glacier melt and on specific mass

balance will be discussed in detail in the results section. Figure 4 shows the resulting quality-controlled and gap-filled data time series of the meteorological variables. All R programming scripts on data processing described here are available upon request.

### 3.1.2 Meteorological observations at Carlini Station

The Argentinean Carlini Station (formerly Jubany Station) is located at $S62°14'$, $W58°40'$ at $15 m a.s.l.$ at a distance of 2.7

km to the AWS. Meteorological observations are carried out by the National Meteorological Service of Argentina (SMN). The data used here represent the time period from 01.01.2001 until 01.01.2016, and contain 3-hourly observations of surface air temperature, wind direction and velocity, barometric pressure as well as cloudiness (SMN, 2016). This time series was re-sampled to hourly data and used to investigate lapse rates between the AWS and Carlini station, in order to gap fill the time series at the AWS during power outages. Details will be discussed in the following sections on meteorological post-processing

and gap filling.

### 3.1.3 Long-term climate dataset of meteorological observations at Bellingshausen Station

The Russian Bellingshausen Station is located on Fildes Peninsula of KGI at $S62°12$, $W58°58$ at about $14 m a.s.l.$ at a distance of 18.5 km to the AWS. The Bellingshausen climate observations are on 6-hourly measurements of barometric pressure, surface air temperature, dew point temperature, relative humidity, total cloud and low cloud cover, surface wind direction, surface wind

speed and precipitation. This time series starts in October 1968 and it is available in 6-hourly resolution (Martianov and Rakusa-Suszczewski, 1989). The time period for the analysis of this paper is from November 2010 to December 2015. The data used here was taken from (AARI, 2016) and was downloaded from the weather data center (weatherdatacenter.com). This data set was used to reconstruct the precipitation time series at the AWS, since the ultrasonic sensor to measure distance to the surface was malfunctioning after less than a year.





## 3.2 Meteorological data processing and gap filling

For driving the glaciological surface mass balance model (GMM) developed by Hock and Holmgren (2005) and Reijmer and Hock (2008), it is imperative to use continuous and quality controlled meteorological input data, i.e. data that have no gaps in the time series and no unrealistic values either caused by instrument malfunctioning or by environmental impacts such as hoar frost. During winter time, power outages are more likely to occur and maintenance is often prevented by unfavorable weather conditions. In order to achieve a 5-year continuous, quality-checked time series for the AWS location on the glacier of the Potter Peninsula, all quality control and gap filling routines were programmed in R language (R Core Team, 2014).

The GMM was run in energy balance mode and the requested climatological input consists of air temperature ($\theta_{air}$), relative humidity ($rh$), wind velocity ($v$), shortwave downwelling radiation ($K_d$), precipitation ($P$), shortwave up-welling radiation ($K_u$), net radiation ($R_n$), long-wave radiation emitted by the earth's surface ($E$), atmospheric long-wave radiation ($A$), barometric surface air pressure ($p_{air}$), albedo ($\alpha$), ice temperature at 5 m depth ($\theta_{ice}$), and cloud cover in eighth's ($c8$).

Gap filling with the monthly mean diurnal cycle was rejected as a method, since a) data gaps in winter time are by far more frequent and also the time period of missing data longer, and b) the diurnal variability is dominated by the seasonal course. In order to produce a gap-filled and quality controlled meteorological time series, at first a continuous time stamp was produced. Then for each sensor, the data were checked for malfunctioning or other environmental impact, but also for statistical properties according to Falk et al. (2016). Each observed meteorological variable was aggregated to hourly time steps to reduce computation time of the GMM. The data set from the Carlini weather station available in 3-hourly (SMN, 2016) data steps was re-sampled to hourly data steps by linear interpolation between the data points and applying a moving average with a width of 24 hours to smooth the edges. Meteorological observations of precipitation and cloud cover at Bellingshausen were re-sampled to hourly data accordingly, to create an artificial diurnal course.

The AWS *air temperature* were screened for spikes (every value outside the six-fold standard deviation of the long-term average) and air temperature readings below $-40°C$ were discarded. The resulting gaps (11.6% of the data set) were filled with records of the UTL air temperature and where not available extrapolated from Carlini (CAR) base air temperature observations based on the monthly mean adiabatic lapse rate analysis carried out by Falk et al. (2016). The remaining missing values of smaller gaps were eliminated by applying a moving average with a small window frame of 3 hours. The configuration of the GMM in energy balance mode allows for the specification of average monthly *air temperature lapse rates*. These were taken from Falk et al. (2016).

Apart from a crossing of a frontal synoptic system, sudden spikes to low values in the *barometric pressure* are usually associated with problems and sudden drops in the power supply. All values below 930 hPa were were thus disregarded. The five-year average of the barometric pressure sensor amounts to 967 hPa and a maximum value of 1009 hPa, which was assumed as maximum value for derived values for the barometric pressure. Data gaps (25% of the data set) were then filled by extrapolating the meteorological observations at Carlini station by applying the hydrostatic equation using the gap-filled air temperature data:

$$p_{AWS} = p_{CAR} \cdot e^{\frac{-9.83 \Delta z}{R_L T}} \tag{1}$$





where the barometric pressure was taken from the re-sampled time series at Carlini base station, $\Delta z$ the elevation difference, $R_L$ the specific gas constant, and $T$ is the absolute air temperature in Kelvin.

The *2m-wind velocity* was screened for sensor malfunctioning. The cleaned wind velocity statistics over five year show a linear relationship between the horizontal wind speeds measured at Carlini station and by the AWS on the glacier of

$$v_{AWS} = 1.15 \cdot v_{CAR} - 2ms^{-1}, R^2 = 0.6. \tag{2}$$

This relation was used to fill the gaps in the time series of the AWS wind speed observations (25% of the data set).

The search of a similar relationship for the *relative humidity* yielded a linear regression between the time series at Carlini base and at the AWS with a very poor correlation coefficient of $R^2 = 0.3$. The comparison of first-order statistics shows that

$$rh_{AWS} = 1.08 \cdot rh_{CAR} \pm 0.5. \tag{3}$$

Generally, the value range of relative humidity is between 60% and 100%. The diurnal variability is usually of the same magnitude ($\sigma rh_{AWS} \approx 0.4$). For this value range the error is thus acceptable. Values for the relative humidity are generally very high at the AWS site. 25% of the complete data set of relative humiditiy were replaced.

*Ice temperature* measurements were taken from the sensor that was originally installed at 10 m depth in November 2010. Due to the extensive ablation over the years, the sensor depth changed significantly over time. The 5-years statistics of the AWS data sets shows clearly that ice temperatures do not drop below $-5°C$. The near-surface levels of ice temperature sensors show values lower than this minimum value, also higher than $0°C$, but this is due to direct contact of the sensor with either melt water or air, hence these periods are excluded from the measurements. Figure 4 shows the course of ice temperature over 5 years in the lowest level, and although the sensor is still in the ice, the fluctuations of the time series reveal that during melt periods, the sensor was partially in contact with either melt water or air. This affected 8% of the data set.

The *cloud cover* data was taken from the observations (cloud coverage $c_8$ in eighth's of total sky area) of the Argentinean Meteorological Service (SMN, 2016). The time series was sampled to 1 hour by interpolation of the 3-hourly data time series. A moving average of 48 h was applied to smooth the resulting time series. This window size is well adjusted not to eliminate the synoptic changes between low pressure systems usually associated with overcast sky and high pressure systems over continental Antarctica often accompanied with clear sky. The time scale of these changes are considered to be at least 3 days to one or two weeks. The cloudiness $c$ is given in decimal format of $c_8$ with a value range between 0 and 1.

The four-component *radiation* sensor is prone to icing or riming due to advection of warm and moist air masses into the region. This affects especially the upward looking sensors for long- and shortwave radiation. As criteria for detection of sensor riming, van den Broeke et al. (2004) suggest that the downward longwave radiation flux density equals the upward flux. Here, the criterion was chosen as

$$LW \downarrow = |LW \uparrow| \pm 0.5Wm^2, \tag{4}$$

where $LW \downarrow$ and $LW \uparrow$ are the upward and downward longwave radiation flux densities, respectively. Additionally, values of $R_n < -500Wm^{-2}$ were discarded. The above criterion was met on less than 1% of the observations. Since air and surface



temperatures are often around melting conditions together with overcast skies or high cloud coverage, the applied criterion was deliberately chosen as very sharp to avoid also filtering real values for longwave radiation flux densities. To fill the gaps in the radiation data time series, the different up- and downward flux densities of the long- and shortwave components were simulated by applying basic geographic and astronomic equations (Campbell and Norman, 2000) using the Julian Day ($J$),

local time ($t$), location information (longitude $\theta$ and latitude $\phi$) in decimal-degree, absolute surface air temperature ($T_a$) in $K$, barometric pressure ($P_a$) in kPa, absolute surface temperature ($T_s$) derived from ice temperatures and upward longwave radiation measurements, cloudiness in decimal numbers ($c$), and albedo measurements ($\alpha$) that were gap-filled (33% ) by linear interpolation between data points and smoothed by a moving average of 48 hours.

The optical air mass number is given for $\psi < 80°$ by $m = P_a/(99 \cdot \psi)$, where $\psi$ is the solar zenith angle. The atmospheric

transmittance $\tau$ is calculated by adapting Gates et al. (1980) to $\tau = 0.5 + 0.45(1-c)$. The top-of-atmosphere (TOA) solar incidental radiation flux ($SW_{TOA}$) is then computed by $SW_{TOA} = 1367 \cdot \cos\psi$, the direct solar incidental radiation flux to the surface ($SW_{direct}$) by

$$SW_{direct} = SW_{TOA} \cdot \tau \tag{5}$$

and the diffuse solar incidental radiation by

$$SW_{diffuse} = 0.4 \cdot (1-\tau) \cdot SW_{TOA}. \tag{6}$$

The solar radiation reflected at the surface is then calculated by multiplying the albedo to the simulated downward shortwave flux which is the sum of diffuse and direct radiation flux. The downward component of the longwave radiation budget at the surface is calculated using the emissivity of the atmosphere ($\varepsilon = 9.2 \cdot T_a^2 \cdot 10^{-6}$) (Monteith and Unsworth, 1990) to estimate the emissivity under cloud coverage ($\varepsilon_{ac}$)

$$\varepsilon_{ac} = (1 - 0.84 \cdot c) \cdot \varepsilon + 0.84 \cdot c. \tag{7}$$

Applying the Boltzmann law this gives for the longwave downward component ($LW \downarrow$)

$$LW \downarrow = \varepsilon_{ac} \cdot 5.67 \cdot 10^{-8} \cdot T_a^4 \, \text{W} \; \text{m}^{-2} \, \text{K}^{-4} \tag{8}$$

and for the longwave upward component ($LW \uparrow$) with a surface emissivity of $\varepsilon = 0.9$

$$LW \uparrow = 0.9 \cdot 5.67 \cdot 10^{-8} \cdot T_a^4 \, \text{W} \; \text{m}^{-2} \, \text{K}^{-4}. \tag{9}$$

The statistics of the simulated radiation fluxes are then compared to the statistics of the measured time series resulting in correlation coefficients generally above $R^2 > 0.6$. The simulated longwave fluxes were adjusted to the observations by fixating the mean values of the simulation to the observations for the longwave radiation fluxes. A difference in long-term average of 29 W/m$^2$ and 35.5 W/m$^2$ was added to the simulated atmospheric longwave emittance and simulated earth's longwave emittance, respectively. The comparison of simulated shortwave fluxes to the long-term observations suggested a further refinement of the

impact of cloud coverage on the shortwave radiation fluxes according to

$$K_{up/down} = K_{up/down,sim} \cdot (1 - 0.4 \cdot c) \cdot 1.171. \tag{10}$$



The overall comparison of the simulated ($sim$) and observed ($obs$) net radiation flux resulted in

$$R_{n,sim} = 15.9 + 0.987 \cdot R_{n,obs}, R^2 = 0.7 \tag{11}$$

a linear regression with very good correlation. Smaller data gaps (less than 24 hours) in the albedo deducted from the shortwave radiation observations were closed by applying linear interpolation and a moving average with a window of 48 hours. The remaining data gaps were closed using the simulated and fitted shortwave radiation fluxes. In summary, about 50% of the shortwave and upwards facing radiation data needed to be gap-filled with the simulated data, whereas only 25% of the earth's longwave radiation data were identified as missing or flawed data.

*Accumulation* measurements based on sonic ranging are available during November 2010 to May 2011 and March 2012 to November 2012. Outside these periods, accumulation was reconstructed using the readings of the Bellingshausen 6-hourly precipitation data. These were aggregated to hourly data time series by linear interpolation and normalization with the total daily sum. The mass balance stake data at the AWS location was then used as an envelop for the daily sums of the resulting accumulation time series.

### 3.3 Mass balance data

Two transects of mass balance stakes (MBS) were installed from the top of the Warszawa Ice Dome down to the border of the glaciers Fourcade and Polar Club to serve for calibration and validation of modeling efforts (see Fig. 1 and 7). Additional transects were installed along glacier ridge on Barton Peninsula on the Fourcade Glacier on the opposite side of Potter Inlet. The stakes were measured at the beginning and end of each summer field campaign in November 2010, February - March 2011, January - March 2012, and every 10 to 14 days, depending on weather conditions, during the austral winter 2012 up to March 2013. During the austral winter 2013 and until May 2016 the measurements were conducted every 20 to 30 days. MBS readings during winter months, were mostly conducted in the ablation area of the glacier except for four stakes (PG04, PG14, PG05 and PG15). Additional measurements in the accumulation area were carried out during summer and fall seasons. The high wind speeds, high precipitation rates during austral winter and potentially the material of the stakes (aluminum) resulted in regular loss of the stakes in the accumulation zone of the Warszawa Icefield and thereby the loss of the time series. For the measurements of stakes that were not protruding vertically from the ice but at an inclination caused by the high wind speeds, a geometric correction was applied to yield the correct exposition length. During the summer field campaigns November 2010 to March 2013, repeat measurements with differential GPS (DGPS) at static points of the mass balance stakes yield an average velocity for the lower transect up to elevations of approx. 250 m of below 1 m a$^{-1}$ and up to 6.3 m a$^{-1}$ for the MBS's on the upper glacier. Figure 7 shows a sample at the MBS transect. The snow density samples were taken with an aluminum snow density cutter (SnowHydro, Fairbanks, Alaska) with a defined volume of 0.001 m$^3$ and a balance scale with an accuracy of 0.1 g (Carl Roth GmbH, Germany) calibrated at least once each summer campaign. For the snow density sampling, a snow pit of about 0.5 m depth was dug out and at least three sample volumes extracted at a depth of about 0.3 m by putting the snow cutter horizontally into the pit wall that were then averaged. The snow depth was measured with a regular snow sonde used as mountaineering equipment. Around each stake about 10 measurements were taken and then averaged. Snow densities were





measured 15 November 2010 in the glacier ablation zone and on 07 March 2011 in the glacier accumulation zone, yielding a snow density of $\rho_s = 503 \pm 16$ kg/ m$^3$ and $\rho_s = 488 \pm 20$ kg/m$^3$, respectively. On 23 January 2012, a snow pit was dug out in the accumulation zone of the Warszawa ice cap at the additional AWS (Falk et al., 2016). Snow density measurements were taken every 0.2 m up to a depth of 2 m, and resulted in an average value of $\rho_s = 416 \pm 47$ kg/m$^3$. The correlation with

depth was not significant. Starting in June 2012 until February 2016, regular snow density values were taken together with the mass balance stake transect measurements. A seasonality in the snow density time series is evident and presented in Fig. 5. With rising temperatures during the austral summer and fall, the snow density also shows rather high values up to approx. 500 kg /m$^3$. With the onset of the winter, the precipitation changes to low density snow, which then over the course of the winter compacts. During austral winter, the overwintering scientist was responsible for the glaciological measurements of the whole

campaign. The snow density time series of the winter 2015 shows a high variability and particularly low values, well below any observation before. Although the values by themselves are reasonable for very cold and dry conditions (Patterson, 1980), the contemporary surface air temperature measurements do not show any behavior that would satisfy the requirements for the very low snow density observation. We therefore suspect the measurement to be affected by errors leading to an underestimation of the snow density. This seems to have been the case during winter 2015. Thus, for the time period before June 2012 and

after March 2015, mass balance and related variables were replaced by monthly averages of the period in between. Figure 5 shows measured elevation profiles of snow density for two dates, i.e. during summer and late fall. The upper profiles show less variability and no significant change for the accumulation area above 250 m, whereas for the lower transect the absolute value of snow density and the variability of the observations is considerably higher. These findings are supported by the snow depth measurements ($D_S$) displayed in Fig. 6, which show the transects *PG0x* and *PG1x* where x is the index of the

actual stake number. The snow depth measurements were carried out as far as possible at the beginning and the end of the summer campaigns, and since 2012 on a more regular basis together with the accumuation and ablation measurements of the MBS's. The lowest MBS of both transects show a higher snow accumulation than the stakes at higher elevation. A reasonable conclusion is a different atmospheric turbulence regime that leads to a higher snow deposition at the glacier border due to the vicinity of the glacier end moraine and moraine. Also in the snow depth measurement, the more distinct stratification of transect

*PG0x* with elevation is evident. The snow density and snow depth time series were used to compute the cumulative ablation and accumulation in m of water equivalent (m w.e.) at the mass balance transects shown in Fig. 8 and 9. To convert to m w.e., values needed to be divided by the density of water at standard conditions $\rho_w = 999.972$ kg/m$^3$. The mass balance stake readings were referred to the initial value measured when the stake was installed, which was later considered as the zero for each stake. The resulting MBS time series graphs is differentiated by the transect ID *PG0* and *PG1*. The gradual shift from ablation at the

lowest MBS PG09 to accumulation at the highest MBS PG04 is clearly visible. The *PG1*-transect on the other hand does not follow this behavior: there is considerably more accumulation at MBS PG19 than at the higher elevation PG17. Also within the accumulation zone, the expected increase of accumulation with elevation does not apply: the cumulative accumulation at MBS PG14 is considerably lower than at MBS PG15. This can be explained by the different degree of exposure to weather. KGI is prone to transient low pressure systems connected to storm events with high wind speeds from Northwest and precipitation

mostly in the form of rain, but also to katabatic winds due to influence of the Antarctic high pressure systems with high wind



speeds from the Southeast (Falk and Sala, 2015). The southern MBS transect *PG1* is more exposed to these synoptic changes
and prone to snow drift by the high wind speeds. The MBS PG19 and partly PG18 show this extra accumulation of the snow
but also PG14 shows a lower accumulation than PG15 although elevations are reversed (see Fig. 9). For the MBS transect *PG1*
it can be assumed that these include effects of aeolic snow drift. The very high temporal resolution of mass balance observation
is unique for the AP region and was chosen to resolve the high seasonal and interannual variability of the onset of accumulation
and ablation period, but also to capture winter melt periods to estimate glacier melt water run-off also during winter time. The
time series analysed here encompass 6 years, but are still ongoing. Figures 5, 6, 8 and 9 show the respective time series of snow
depth, snow density and surface mass balance observations. The glaciological observations reflect the heterogeneous pattern
of accumulation and ablation areas reported by Falk et al. (2016). Figures 8 and 9 include the spread of the GMM output of
accumulation and ablation for the DEM pixels at MBS locations PG05 and PG08 (Fig. 8), and PG15 and PG18 (Fig. 9). The
GMM output for the stake locations close to the glacier border, i.e. PG09 and PG19, were excluded in the graphs, as already
discussed in this section. The GMM outputs the cumulative accumulation and ablation for 10 m by 10 m grid cells at the stake
locations. Model calculations are shaded in grey. Figures 8 and 9 show that model and observations are generally in agreement.
The GMM does not fully account for the spread in the observations which is attributed to the significant snow drift due to
high wind speeds and snow deposit due to turbulence at the glacier end moraine, and to the high contribution of snow erosion
by rain. The validation of GMM results and comparison to observations are subject of section 3.5. The results of model and
observations will be further discussed by specific glacier surface mass balance in sections 4.1).

## 3.4 Hydrological catchment definition of Potter Cove and input grids to the glaciological model

To estimate the complete input of glacial and snow melt water into the Potter Cove, the GMM was run in catchment configu-
ration. The model area encompasses glacial and periglacial areas that are part of the Fourcade Glacier catchment area draining
into Potter Cove. The boundary of the Fourcade glacier taken from the Bishop et al. (2004) and refined with the analysis of
glacial divides by own kinematic differential GPS mapping of surface elevation. The surface topography is based on an analysis
of TerraSAR Tandem-X remote sensing data performed by Braun et al. (2016) with a resolution of this DTM is 10 m by 10
m. The GMM was thus applied to two different catchment areas: 1.) the hydrological catchment area draining into the Potter
melt water and discharge creeks, and 2.) the Potter cove catchment, i.e. the Fourcade Glacier, defined as the glacier area of the
Warszawa Icefield that drains into Potter Cove. The Potter Cove catchment has an area of 25.1 km$^2$ and is glacierized to 94%
where the greatest non-glacierized part is located on Potter Peninsula. The map in Fig. 1 displays the catchment definitions,
the location of MBS transects in the catchment area. The input data grids to the GMM are based on this DTM. The DTM
published by Rückamp et al. (2011) has a spatial resolution of only 50 m by 50 m but its accuracy is rather low especially for
the Warszawa Icefield due to missing *in-situ* ground truth data. Apart from the DTM and the catchment area definition, further
input grids comprising information on slope, aspect and sky view factor were calculated from the DTM. The grids containing
the information of glacier facies, i.e. firn, ice and rock area, were derived from the glacier zonal mapping published by Falk
et al. (2016). The grid containing initial snow water equivalent values in cm were taken from own in-situ measurements along
the MBS transects and spatially extrapolated by using the elevation from the DTM. The inaccuracies of the different grids on





information of glacier facies and initial snow height, are taken into account by cutting off the first 1056 hourly data time steps (equivalent to 44 days) of the model run. During this initialization period, the GMM adjusts the inaccurate spatial input data. After these initial 1056 hourly time steps the model-internal physics are assumed to be according to the actual state of physics of the glacier under investigation and a realistic discharge pattern established.

For the catchment boundary refinements, flow directions were calculated on the basis of the data from own differential GPS measurements on the MBS transects taken at the glacial surface and then were interpolated to form a topography of the surface. For the austral summer 2010-2011, the drainage basins were estimated to encompass glacier elevations between 80-450 m with slope to the SW. The supra-interglacial drainage pattern analysis using water drainage channels on the glacier surface were identified using sensors in optical remote sensing satellite data (SPOT-4, 18 November 2010, ©ESA TPM, 2010).

The drainage of the mass of a glacier can be interpreted similar to a karst rock (Eraso and Domínguez, 2007). The Fourcade glacier surface drainage shows a straight, poorly integrated and strong direction towards the Southwest. The glacier divides then served for further refinement of the catchment area definition in especially of the southern part of the Potter Cove catchment (see Fig. 1).

### 3.5   Glaciological model calibration

The glacier surface mass balance model (GMM) by (Reijmer and Hock, 2008) computes accumulation and ablation, including glacier melt and discharge, at hourly resolution. The GMM is fully distributed, meaning that calculations of glacier surface mass and energy balance terms are performed for each grid cell of a defined model area on a digital elevation model as discussed in the last paragraph. Discharge is calculated from the water provided by melt plus liquid precipitation by three linear reservoirs corresponding to the different storage properties of firn, snow and glacier ice volumes. The energy available for melt ($Q_M$) is

computed by

$$Q_M = G(1 - \alpha) + L_{net} + H + \lambda E + Q_{ground} + Q_R, \tag{12}$$

where $G$ is the global (solar incidental radiation) radiation, $H$ the sensible heat flux, $\lambda E$ the latent heat flux, $Q_{ground}$ the ground or ice heat flux and $Q_R$ the sensible heat supplied by rain. Global radiation, albedo, as well as up- and downwelling longwave radiation were taken from the AWS time series. The roughness length for wind were set in the GMM configuration and used

to tune the model to the mass balance stake observations at the transect *PG0x*. In the configuration of Braun and Hock (2004) for a summer period on the little Bellingshausen ice dome during a weeks during austral summer, a value for $z_0 = 0.0026$ was chosen. This choice proved inadequate for the wider region of the Warszawa Icefield. Smeets and Van den Broeke (2008) and Heinemann and Falk (2002) found an aerodynamic roughness length of $z_0 \approx 10^{-4}$ to $10^{-5}$ near the equilibrium line on the Greenland ice sheet. A study on roughness lengths and parametrizations of sensible and latent heat fluxes in the atmospheric

surface layers for alpine glaciers (Brock et al., 2006) suggests a significant lower value for $z_0 \approx 10^{-5}$ m depending on the very high wind speeds and a surface that is during most summer periods characterized by slush and bare ice with water-filled gaps (see photo in Fig. 2). Here, $z_0 = 0.00005$ was chosen. The roughness lengths for temperature and vapor pressure were computed according to Andreas (1987). Lapse rates for air temperature, precipitation and wind are configured in the GMM



input parameters according to Falk and Sala (2015). Air temperature, wind velocity and relative humidity were then calculated on spatial distribution over the whole area. The turbulent fluxes of latent and sensible heat were computed according to Monin-Obukhov similarity theory considering atmospheric stability. The surface temperature is derived by the GMM by iteration from the energy balance equation, where the surface temperature is lowered in case of a negative $Q_M$ until the term becomes zero.

The ground or ice heat flux, otherwise neglected, is thereby considered indirectly: if the energy balance equation is negative, the surface temperature is being increased using negative surplus. No melt is allowed until the negative energy balance has been compensated for. The energy amount supplied by rain is computed by

$$Q_R = c_w R^\star (T_r - T_s),\tag{13}$$

where $T_r$ is the temperature of rain assumed to be identical to surface air temperature.

The GMM run period was s chosen for the 5-year period 22 November 2010 to 21 November 2015. The glaciological observations encompass the time period November 2010 - May 2016, but the high-frequency data acquisition does not start before February 2012. For calibration of the GMM, the mass balance observations at the transect with transect ID $TID = PG0$ (see Fig. 10), and for validation the MBS transect with $TID = PG1$ (see Fig. 11) were used. In both figures, each start of the ablation period was chosen as null reference for the cumulative mass balance to highlight two main drivers of the GMM

deviation from observations, defined as the summer year (SY). The errors are cumulative over the course of each year starting with the ablation period (e.g. SY=2012 starts 02 Dec 2011, Table 1). Figure 10 shows the high interannual variability. The high frequency of observations allows to differentiate between periods with different climatic settings. The high deviation of cumulative mass balance in especially towards the end of the ablation period show that the GMM clearly underestimates the accumulation due to snow drift and turbulence-driven snow deposition at the glacier border (see Fig. 10 and 11). The closer the

MBS are to the glacier border, the more the effect of turbulence-driven snow deposition comes into effect. The year SY=2014 contained the coldest summer in the GMM run period, with the highest amount of precipitation as well. In the beginning of the ablation period, air temperatures were below freezing and snow fall occurred together with high wind speeds. This leads to an underestimation in the simulated cumulative mass balance since the GMM does not account for snow drift. Then there is a period when model physics and observations are in very good agreement. During the fall of SY=2014, there were frequent

rain events. Since the GMM was run in catchment configuration, meaning that the model area contained also the periglacial parts of the hydrological catchment, the snow module (Reijmer and Hock, 2008) could not be applied. Refreezing processes could, thus, not be considered, and the simulated cumulative mass balance is also underestimated. The accumulation period at the end of SY=2014 is again marked with the underestimation in cumulative mass balance due to the disregard of snow drift in the GMM. Apart from these two climatic boundary conditions, the model physics and the glaciological observations are in

good agreement, and a drift or disagreement over the 5-year GMM run period cannot be seen in the data.





## 4   Results

### 4.1   Glacier mass balance

Cogley et al. (2011) give the climatic-basal mass balance as the sum of surface accumulation $c_s$, surface ablation $a_s$, internal accumulation $c_i$, internal ablation $a_i$, basal accumulation $c_b$ and basal ablation $a_b$:

$$b = c_s + a_s + c_i + a_i + c_b + a_b \tag{14}$$

The sum of $c_b$ and $a_b$ is the basal mass balance. The climatic-basal mass balance includes all those components of mass change that do not arise from glacier flow or frontal ablation. In the following, point mass balances are denoted by lowercase letters $b$, whereas glacier-wide mass balances are denoted by uppercase letter $B$. Ice density is assumed to be $\rho_{ice} = 900\,\text{kg/m}^3$ equivalent to 0.9 kg/l. All units are in $\text{kg/m}^2$. Assuming an uncertainty in snow height measurements of $\Delta H = 0.2$ m to account for natural surface heterogeneity around each stake and of $\Delta\rho = 10\,\text{kg/m}^3$ for the variability in snow density measurements, leads to an uncertainty of $\Delta b = 0.35$ m w.e. in specific glacier mass balance estimates.

The GMM outputs for the first glaciological year 2010 (meaning 2010/2011) were excluded from further analysis due to the model spin-up for the first few months (see in section 3.4). Beginning and end of the accumulation and ablation period of each year are listed in Table 1, and show the seasonal shift towards late start of the ablation period until November or December and the late end of the ablation period until end of May, except for the glaciological year 2013/2014. The start of the accumulation period was used for the definition of the glaciological year. Interannual variability is very high and can differ by more than two months in between years. It reflects the variability in the meteorological drivers that is discussed in detail by Falk and Sala (2015).Therefore, the annual mass balances were calculated in sub-stratigraphic system (see Fig. 12 and 13). In the following, we only differentiate between winter (i.e. June to November) and summer period (i.e. December to May).

The glaciological year 2011/2012 contained a very cold and dry winter in 2011. The summer 2012 showed an exceptionally high net radiation balance amounting to 156 % of the seasonal 5-year average. The resulting high ablation entails a strongly negative specific net balance. In July of the winter 2012 brought a two-weeks period of rain together with above freezing air temperatures leading to the erosion of the fresh snow pack, and in the net balance to low accumulation. Although the summer 2013 showed low ablation due to high cloud caverage (less precipitation), the glaciological year 2012/2013 remains in its net balance negative. It was also a year with very low winds during summer and very high winds during winter. The glaciological year 2013/2014 reveals a wet winter with high accumulation rates but also a very cold and very wet summer resulting in low ablation. Regardless of this, the net balance was negative. The glaciological year 2014/2015 started with a warm and wet winter, followed by a warm and dry summer leading to higher ablation rates. 2015/0216 was a very strong El Niño year with a very cold winter in 2015, followed by a warm and very long summer in 2016. The MSB PG19 (at the elevation of ca. 100 m) clearly reflects the effect of turbulence-driven snow accumulation at the glacier end moraine. Estimates of $b_n$ are significantly higher than the GMM output for this stake location. The variability and impact of snow drift and turbulence-driven snow deposition is visible in the locally calculated estimates of $b_n$ of the MBS observations. From the graphs (Fig. 12 and 13), it becomes clear that winter accumulation is in most of the years not sufficient to cover for the summer ablation. Summer ablation or the specific




summer mass balance on the other hand reveal to be highly variable between years, depending on climatic conditions and the length of the ablation period of the respective glaciological year. The equilibrium line altitude (ELA) conveys an instantaneous response to the climatic boundary conditions, and will be discussed in section 5. The accumulation area ratio (AAR) defines the boundaries of the glacier in equilibrium state with its climatic boundary conditions. The expected future extent of the ice

cap of the Fourcade will be discussed in section 6.

### 4.2 Melt water discharge into Potter Cove

The GMM calculates the discharge on an hourly basis according to the temporal resolution of the meteorological input time series. The GMM configuration allows for computation of partly glaciated catchment areas as is the case for the Potter Cove catchment. The GMM differentiates between the source areas of the melt water discharge:

$$q_{sim} = q_{firn} + q_{snow} + q_{ice} + q_{rock} + q_{ground}. \tag{15}$$

Figure 14 shows the temporal evolution of the total melt water discharge from the complete Potter Cove catchment ($q_{sim}$), and the respective source areas (firn, ice, snow and rock). The penetration and discharge through ground is negligible, $q_{ground} \approx 0$. Area sizes change over the course of a year. Figure 14 shows the transitional importance of the dominating source area to glacial discharge throughout the seasons. The contribution of firn areas to glacial discharge is mainly controlled by the seasonal course

of surface air temperature and net radiation balance. Since the albedo for firn areas remain high during the summer ($\alpha > 0.75$), firn surface melt starts in spring and consistently pertains throughout the summer with monthly discharge quantities of $< 0.5$ m$^3$/s. The glaciological year 2014/2015 contained a very cold and moderately wet winter followed by a very warm but dry summer. This resulted in a pronounced time lag between snow and ice area contribution to glacial discharge, but also high melt water quantities due to the amount of winter precipitation. Glacial discharge from snow areas predominate the first part of a

summer season and from ice areas in the second part. Differences in monthly discharge from snow areas can amount to nearly m$^3$/s between years. The high variability of ablation and accumulation reflects the very high inter- and intra-annual variability of the meteorological boundary conditions (Falk and Sala, 2015). The frequent occurrence of melt periods during winter is attributed to advection of moist and warm air masses from mid-latitudes by synoptic low-pressure systems. This results in non-zero discharge for winter season (JJA: June to August). Figure 15 shows the seasonal sums of glacial discharge from Fourcade

Glacier into Potter Cove as sums.

Linearly relating the time series of simulated discharge to the Positive Degree Day ($PDD$) time series derived from the Carlini air temperature series (Falk and Sala, 2015) shows a high correlation coefficient

$$q_{sim} = 0.2 + 0.05 \cdot PDD, \ R^2 = 0.84. \tag{16}$$

This means that 84% of the changes in discharge can be explained statistically to changes in air temperature. Observed tem-

perature trends are highest in winter months, i.e. a trend in minimum air temperature of nearly $5°C$ over 4 decades for August (Falk and Sala, 2015). This might result in lesser accumulation during winter and, thus, in a more negative mass balance. The linear regression between the simulated discharge to the PDD time series calculated from the AWS data shows a lesser





correlation

$$q_{sim} = 0.4 + 0.14 \cdot PDD, \; R^2 = 0.74. \tag{17}$$

Above glaciated areas, surface air temperature is being reduced by the melt processes. Thus, PDD and degree day factor analysis are best derived from air temperature observations that are within the catchment area but on non-glaciated areas.

## 5 Discussion of equilibrium line altitude analysis

Figure 16 shows the equilibrium line altitude (ELA) estimates from observations (black) at the transects PG1x (solid circles) and PG0x (open squares), and from the GMM output (red). Error bars are derived from the regressions. The calculated ELA's are generally around 260 m altitude. Curl (1980) states that all glaciers on KGI appear in near-equilibrium and slightly negative conditions and that geological evidence shows no major detectable advance within the past two centuries. Braun and Gossmann (2002) have compiled values of the ELA obtained by different authors in the South Shetland Islands region along the last decades. Jiawen et al. (1995) do not give uncertainties of their glaciological observations and ELA estimates, but state that a high variability between different consecutive years was observed. Jiahong et al. (1994) and Jiahong et al. (1998) concluded from mass balance studies on Collins or Bellingshausen Dome, that this small ice cap was in steady state between 1971 and 1991 in agreement with Curl (1980). Serrano and López-Martínez (2000) presented a concurrent ELA for the South Shetland Islands located around 165 to 250 m a.s.l. Bintanja (1995) asserted that the ELA lay around 100 m in the Ecology Glacier, KGI. Despite the strong variability, the ELA has increased by more than 100 m from the late 60's up to the 90's. More recently Osmanoglu et al. (2014), Navarro et al. (2013), Navarro et al. (2009) and Molina et al. (2007) have placed the ELA at $\sim$230 m and $\sim$ 187 m for Hurd and Johnsons glaciers, respectively, both located at Livingston Island ice cap. Until recently, the remote sensing data from synthetic aperture radar (SAR) measurements only allowed for analysis during the summer months due to seasonal manning of the Chilean base O'Higgins responsible for the download of data from the overpassing satellite, thus only resulting in the estimation of the transient snow line, in this case the firn line. Hence, the ELA would be systematically underestimated during a negative surface mass balance year. Additionally, SAR data was not corrected for incidental angles, thus not allowing for differentiation of ablation patterns from superimposed ice. The ELA analysis here is based on ground-based glaciological studies only, and remote sensing studies have been disregarded to avoid methodological bias in the ELA analysis. Multitemporal SAR data analysis carried out by Falk et al. (2016) resulted in an ELA of ca. 250 m for the glaciological year 2010/2011 during a cold and snowy summer. Field observations support the importance of snow drift on the accumulation patterns due to the high wind speeds and the related snow drift and accumulation according to domes and troughs. The average ELA were derived from own glaciological studies on KGI over the time period 2010-2015 yields an average $ELA = 260 \pm 20$ m.

Figure 17 places the results from own glaciological observations into the long-term context. Where available, error or uncertainty bars were included. The temporal evolution of the ELA estimates from the different glaciological studies show clearly that until the 80's the glaciers on KGI were in near-equilibrium (Curl, 1980), but that after this time period there is a clear





increase in the ELA. Our analysis in the preceeding paragraphs also shows that the accumulation of Fourcade Glacier generally does not suffice to account for the ablation, but also that the interannual variability is high especially during the ablation period. The Fourcade Glacier is clearly in retreating mode, also due to the fact that ice flow velocities in lower glacier on Potter Peninsula are below 1 m/year (Falk et al., 2016). The ice mass flux cannot compensate the losses by discharge in the ablation

zone.

## 6   Conclusions on expected glacier extent by accumulation area ratio

Time series of accumulation and ablation show a very high intra- and interannual variability that concurs with climatological variability reported by Falk and Sala (2015). The observations at two mass balance stake transects demonstrate the high spatial variability, regular occurence of winter melt periods and the impact of snow drift, turbulence-driven snow deposition, snow

layer erosion by rain and high exposition to synoptic impact. These processes are not yet included in the model physics and can lead to discrepancies between GMM simulation and observations under specific climatic conditions. Overall, observations and model are in good agreement though. The high interannual variability in climate conditions, accumulation and ablation patterns is propagated to variability in glacial discharge time series. The difference between years can be as high as 40%. The simulated glacier discharge is highly correlated to PDD time series with a coefficient of determination of $R^2 = 0.84$. One

of the most intuitive parameters to describe the equilibrium state of a glacier is the accumulation area ratio (AAR). Defined as the area of glacier accumulation versus the total glacier area, this parameter describes the state and health of a glacier. Particularly, it is an indicator for expected future retreat or growth of a glacier until it gets into equilibrium with concurrent climatic conditions. It thus relates via the ELA with the specific mass balance and with the possible retreat or growth of the glacier. The more negative the mass balance, the higher the elevation line of ELA and the smaller the AAR. Once the long-

term ELA reaches values higher than the maximum altitude of the glacier dome, there is no more accumulation area and the glacier will disappear sooner or later. Möller and Schneider (2015) define this as the turning or tipping point of the glacier evolution. This behavior also depends on the bedrock topography which remains largely unknown for the Warszawa Icefield. If underneath the glacier is mountainous terrain, then the stratigraphy of glacier ice mass would persist longer due to higher elevation than if it would all be ice mass underneath. Glaciers respond to climatic boundary conditions with a time lag of a

few decades to centuries until they are in equilibrium with the climatic conditions (Paterson, 1994). The already committed mass change will lead to further retreat of the glacier border within the near future. The expected ratio of accumulation area to the total glacierized area for a glacier in equilibrium with its climate is within the range of $AAR_{eq} = 0.5$ to 0.8, e.g. Meier and Post (1962) and Paterson (1994). Corresponding to the observed average $ELA = 260$ m with concurrent glacier extent is a ratio of approx. $AAR = 0.26$ for the Potter Cove catchment, meaning the glacier is clearly retreating until it reaches its

equilibrium state shown in Fig. 15. Here, the yellow line marks the glacier extent for elevations above 110 m referring to an $AAR = 0.8$ and the orange line marks the glacier extent for elevations above 230 m referring to an $AAR = 0.5$. Underlying is the assumption that lower elevations will melt homogeneously and ice velocities are negligible as soon as the glacier terminates on land in the Potter Cove catchment. A tipping point will be reached when the ELA rises above the highest point of the glacier



which implies that the accumulation area goes to zero, i.e. an $ELA > 490$ m. Even if the observed trends in climatic boundary conditions does not continue, the Fourcade glacier would still retreat until it reaches equilibrium with climatic conditions. Its extent will be defined by these conditions and the glacier border located between the yellow and the orange line in Fig. 15. The equilibrium state corresponds to a glacier extent that would approximately end between the elevation lines of $h_{elev} = 110$

m (Fig. 15 yellow line) and $h_{elev} = 230$ m (Fig. 15 orange line) referring to an $AAR_{eq} = 0.8$ and $AAR_{eq} = 0.5$, respectively. Thus, further retreat of the glacier border is to be expected. The southern part of the Fourcade glacier on Potter and on Barton Peninsula contain a major area of very low ice flow velocities $< 1$ m$^2$ and below the ELA in the ablation zone. These areas can be assumed to decrease the most and the glacial melt water streams are likely to significantly increase their sediment freight due to larger distances through moraine landscape.

## 7  Code availability

The R codes are available on request from Ulrike Falk.

## 8  Data availability

Supplementary data are available at: https://doi.org/10.1594/PANGAEA.874599, http://dx.doi.org/10.1594/PANGAEA.848704.

*Author contributions.*  UF was the PI of the glaciological and climatological work package within the IMCOAST project, and also leading

the glaciological modelling work on KGI within the IMCONet project. DL was responsible for the quality assessment and control of mass balance data time series, the establishment of final glaciological protocols and all communication with the overwinterers. All final analysis and post-processing of climatological and glaciological data time series was performed by UF. ASB investigated the hydrology of Potter Cove, refined and implemented the catchment definition grids for the glaciological surface mass balance model. Glaciological modelling work was carried out by UF, with ASB contributing to the calibration and validation procedures. The manuscript was mainly written by UF.

DL contributed to the sections concerning mass balance stake observations and analysis, as well as glacier mass balance. ASB wrote parts for the hydrological catchment definition and contributed to the sections on glacier discharge and melt water analysis

*Competing interests.*  There are no potential conflicts of interest regarding financial, political or other matters.

*Acknowledgements.*  We would like to thank the Alfred Wegener Institute (AWI) from Germany and the Instituto Antártico Argentino - Dirección Nacional del Antártico (IAA-DNA) from Argentina for their support in Antarctica. A special acknowledgement goes to Hernán

Sala for assistance in the field and help with logistics and to the overwintering scientists at Carlini Station and Dallmann Laboratory: Daniel Viqueira, Juan Piscicelli, Facundo Alvarez, Francisco Ferrer, Pablo Saibene, Martín Gingins and Julia Luna without whom this work would not have been possible. During the period 2010-2015, the overwintering crews from the Ejército Argentino at Carlini Station continuously supported our scientific tasks and we would like to specifically mention SP Norberto Leonardo Galván. All graphs in this paper were produced



using the R programming language (Team, 2014) and QGIS software (Team, 2016). We also thank the funding support provided by the Marie Curie Action IRSES (FP7 IRSES, Action No. 318718), the logistic support by the Universities of Bonn, Bremen and Erlangen-Nuremberg and the Argentinean Antarctic Institute (IAA-DNA). We thank Ben Marzeion for the careful reading of and his very helpful comments on of the final manuscript.





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





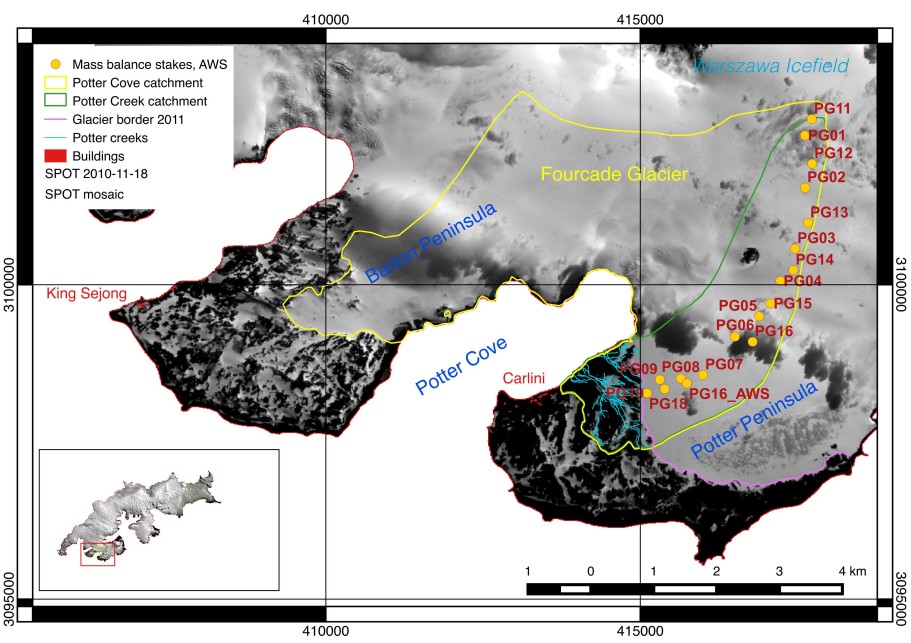

**Figure 1.** Map of research area on King George Island (South Shetland Islands, Northern Antarctic Peninsula) including the locations of own installations and external data time series. Potter creek basins of Potter North and Potter South with drainage channels and mass balance stake locations along two transects, PG0x and PG1x (where x is a placeholder for stake number) in the catchment area. Background: SPOT-4, 18 November 2010, ©ESA TPM, 2010




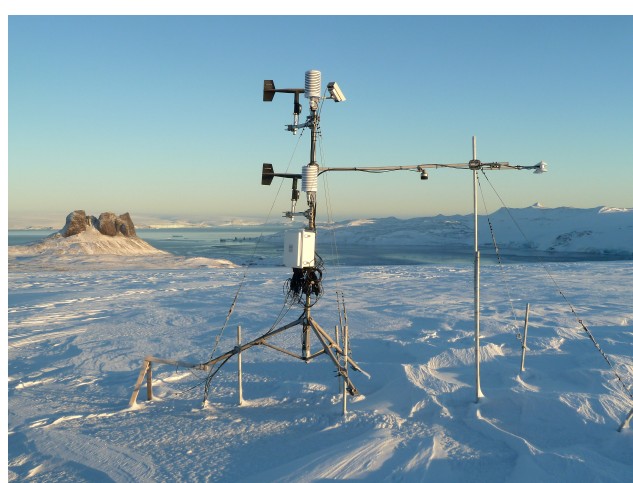

**Figure 2.** AWS installed on the Fourcade glacier with view to the Potter Cove and Three-Brother Hill. The photo was taken during winter on 30 May 2012.

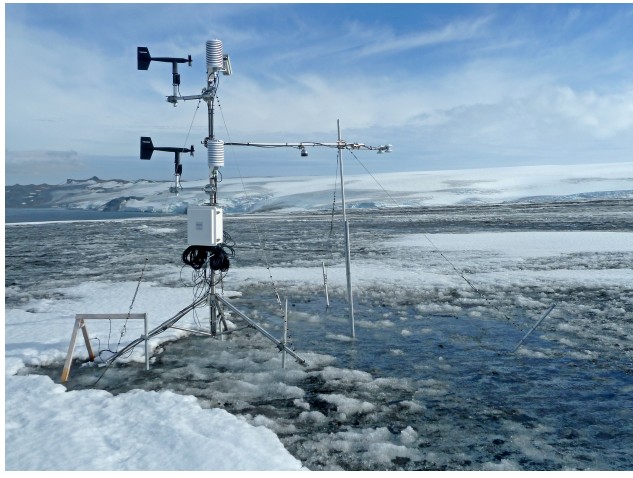

**Figure 3.** AWS installed on the Fourcade glacier with view to the Potter Cove and Three-Brother Hill. The photo was taken on 04 March 2012 and shows the AWS during the ablation period when pyroclastic material resurface due to melting of the winter snow layer.



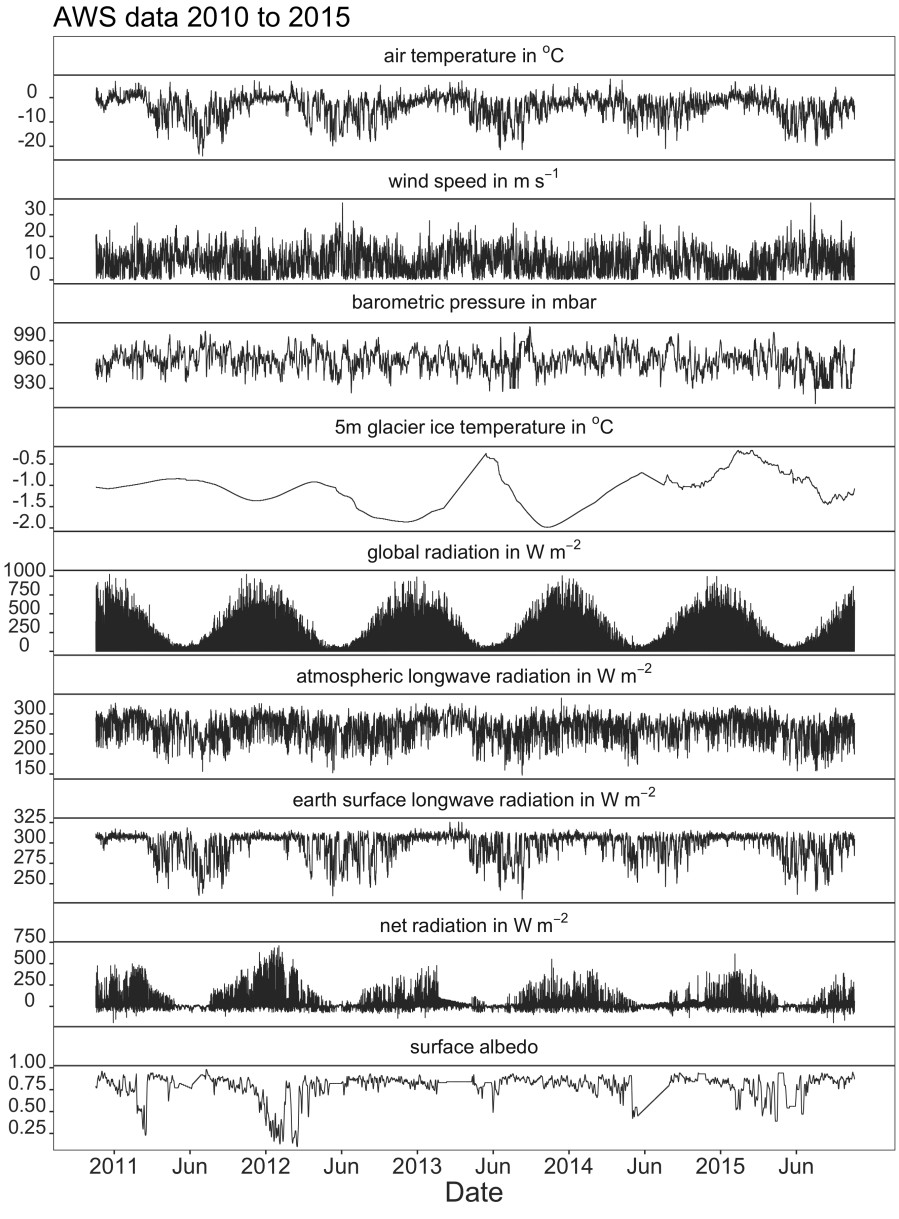

**Figure 4.** Meteorological time series (gap-filled) aggregated to hourly resolution at the AWS site on the Fourcade Glacier, King George Island, during the time period November 2010 - November 2015.



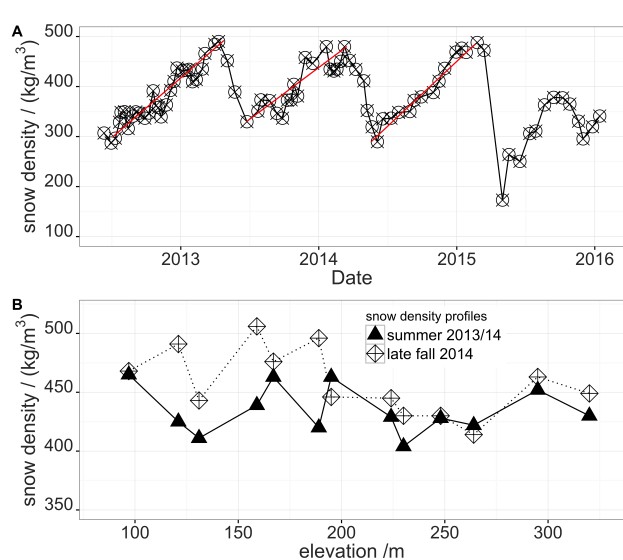

**Figure 5.** Displayed is the time series of snow density observations on the Fourcade Glacier, Potter Peninsula, during the time period June 2012 to February 2016 (A). Red lines are linear regression lines from onset of winter accumulation until the end of the glaciological year. The snow density profile measurements along the mass balance stake transects (see Fig. 1) on the dates 21-January-2014 (summer) and 10-March-2014 (late fall) on the Fourcade Glacier, Potter Peninsula, are shown (B). Measurements were taken at a depth of 30 cm.





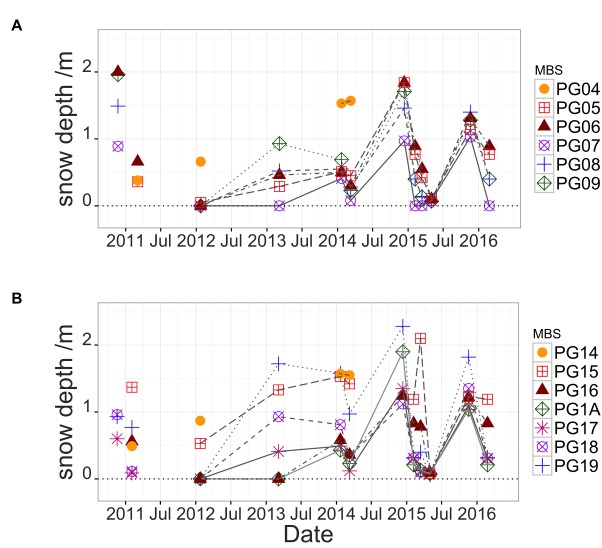

**Figure 6.** (A) Time series of snow depth measured at the two mass balance stake transects (PG0x and PG1x) on the Fourcade Glacier, Potter Peninsula, during the time period November 2011 to May 2016. (B) Snow depth profile with elevation measured during summer and late fall 21-01-2014 and 10-03-2014. The location of the individual mass balance stakes (MBS) is shown in Fig. 1.





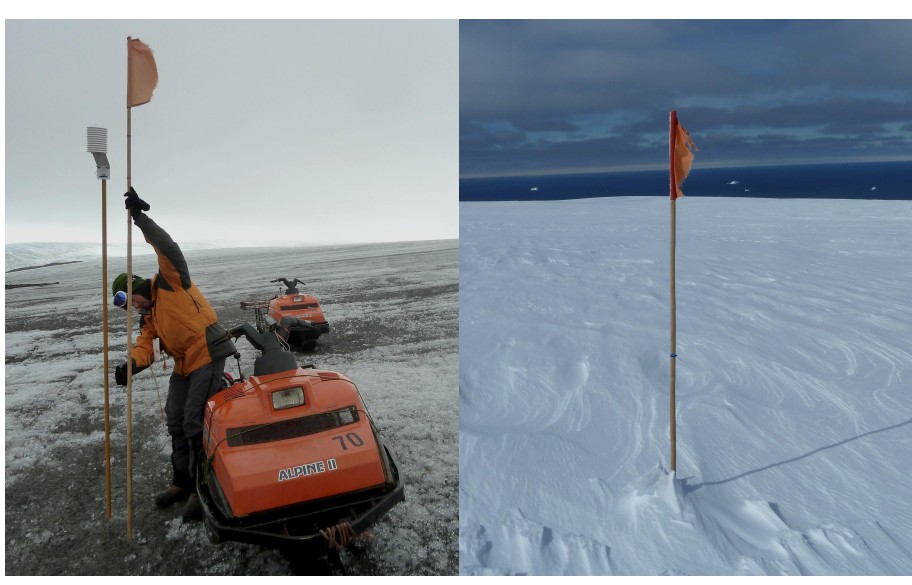

**Figure 7.** MBS transects installed on the Fourcade glacier during the ablation period (left) and during the accumulation period (right) in 2012.





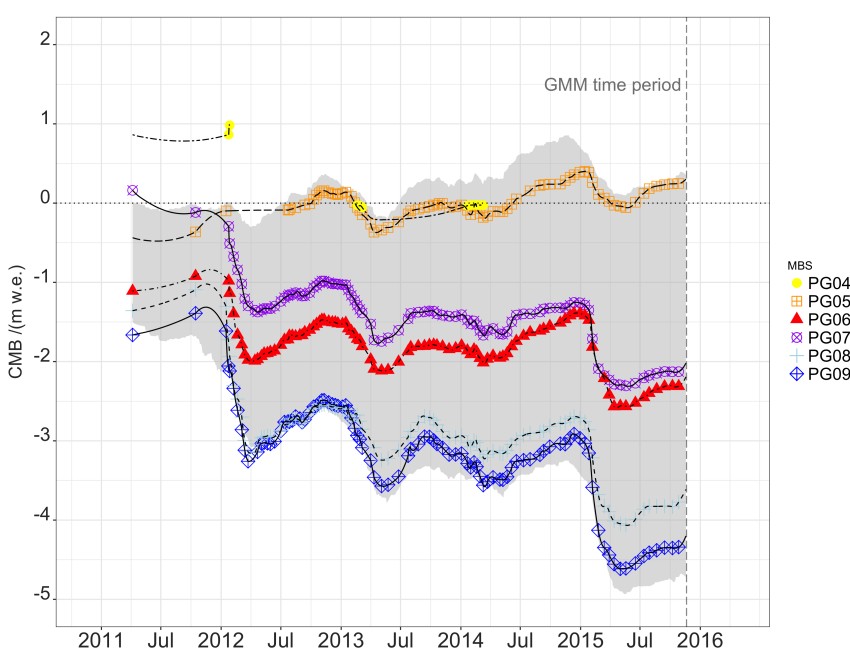

**Figure 8.** Shown is the cumulative mass balance in $mw.e.$ (water equivalent) measured at the calibration mass balance stake transect $PG0x$ on the Fourcade Glacier, Potter Peninsula, during the time period November 2011 to May 2016. The location of the individual stakes is shown in Fig. 1. The grey shade indicates the spread of the GMM-simulated cumulative mass balance at the stake locations PG05 to PG08.





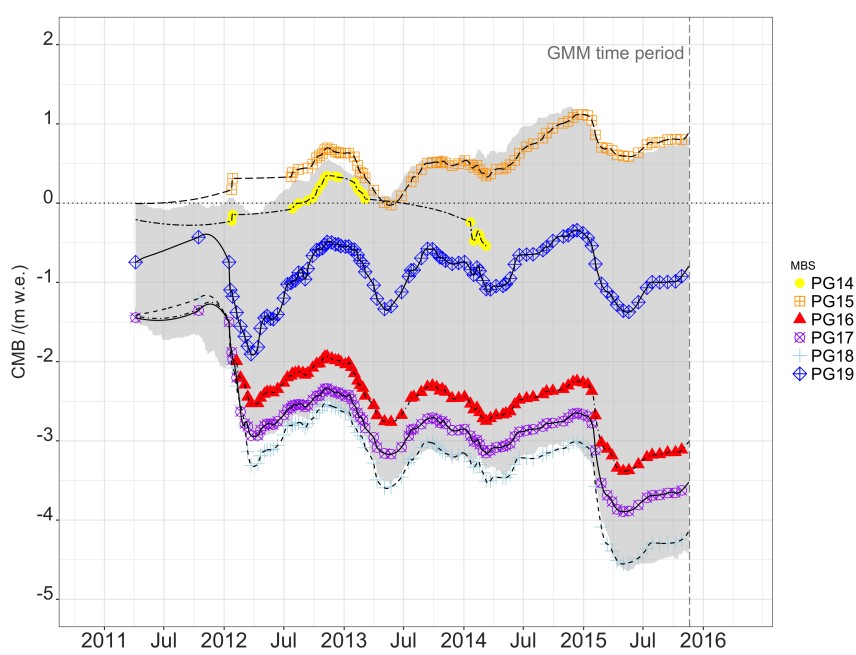

**Figure 9.** Shown is the cumulative mass balance in $mw.e.$ (water equivalent) measured at the validation mass balance stake transect $PG1x$ on the Fourcade Glacier, Potter Peninsula, during the time period November 2011 to May 2016. The location of the individual stakes is shown in Fig. 1. The grey shade indicates the spread of the GMM-simulated cumulative mass balance at the stake locations PG15 to PG18.




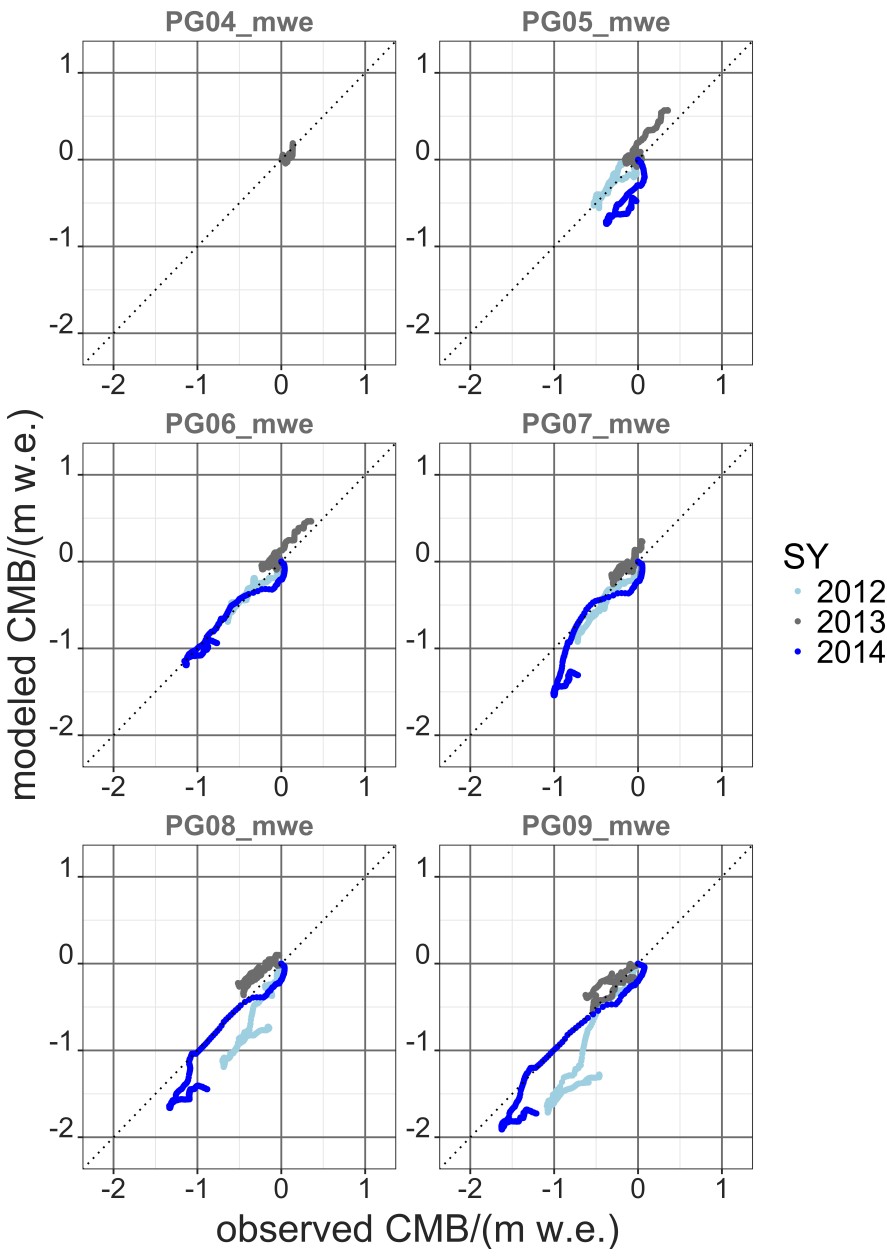

**Figure 10.** Specific surface mass balance from simulations with the glacier melt model (mod) and observation on the Warszawa Icefield (transect ID, TID: PG0x) for calibration purposes. Year numbers give the year of start of the glaciological mass balance year. It encompasses the glaciological years 2012/2013 to 2014/2015.



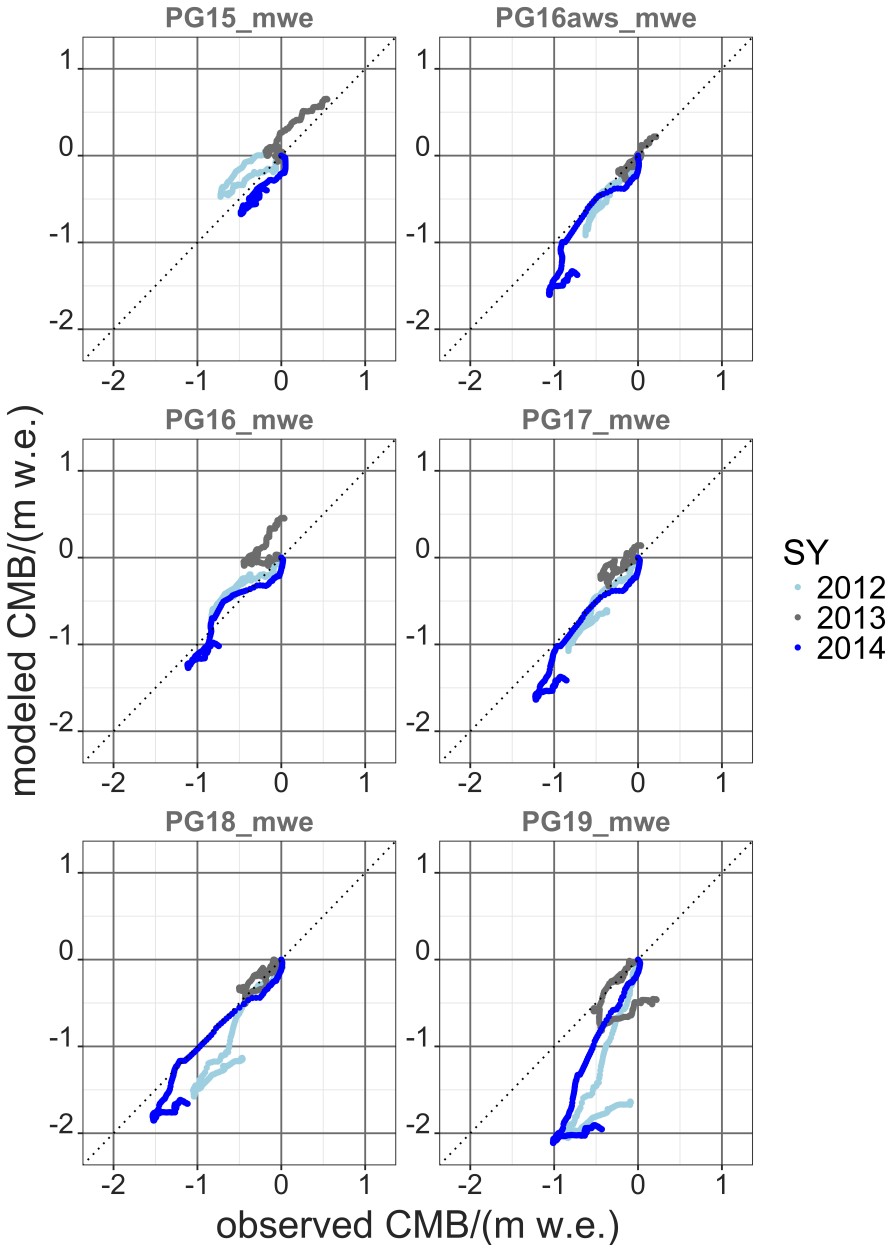

**Figure 11.** Specific surface mass balance from simulations with the glacier melt model (mod) and observation on the Warszawa Icefield (transect ID, TID: PG1x) for validation purposes. Year numbers give the year of start of the glaciological mass balance year. It encompasses the glaciological years 2012/2013 to 2014/2015.





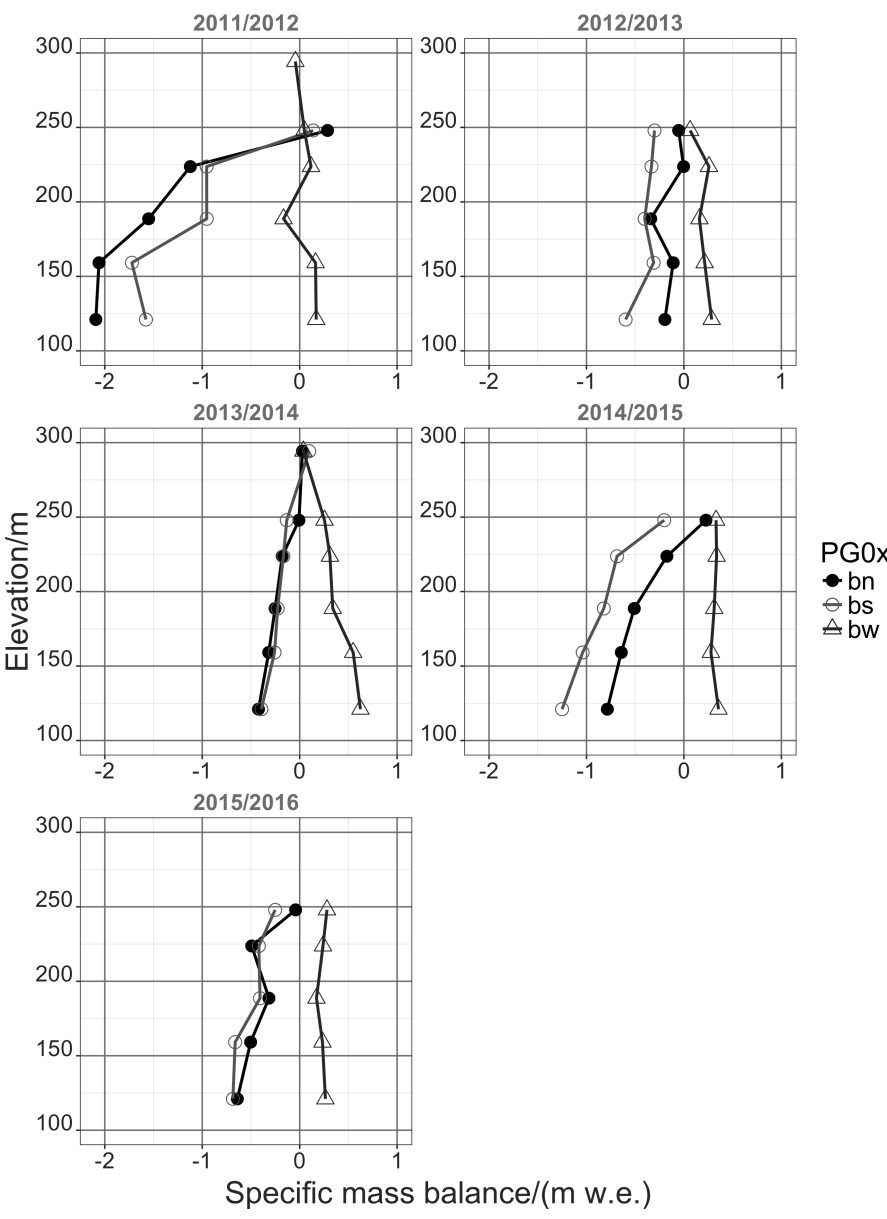

**Figure 12.** Specific summer, winter and net mass balance ($b_s$, $b_n$ and $b_n$, respectively) derived from mass balance stakes observation at transects PG0x during 2010 to 2016.



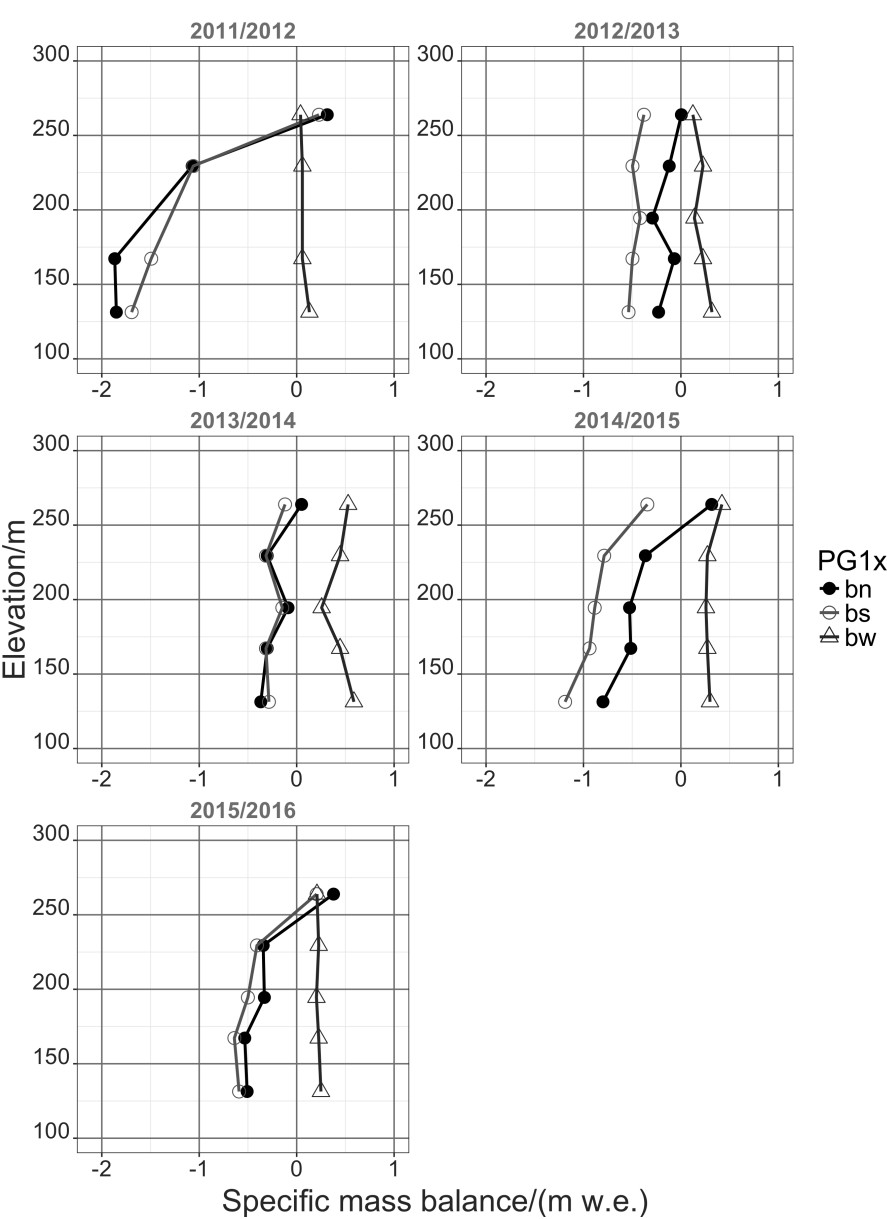

**Figure 13.** Specific summer, winter and net mass balance ($b_s$, $b_w$ and $b_n$, respectively) derived from mass balance stakes observation at transect PG1x during 2010 to 2016.





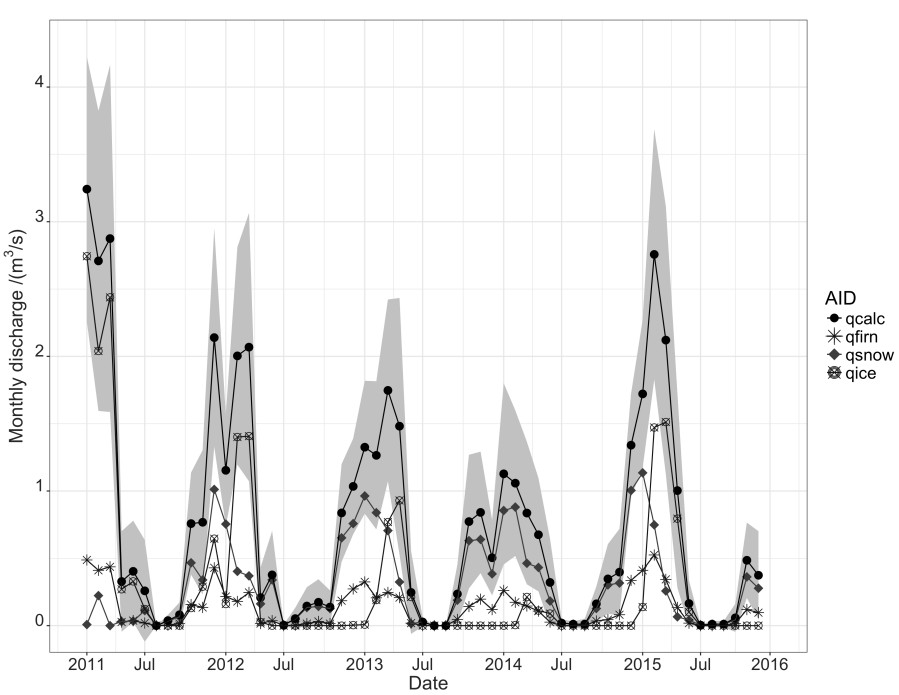

**Figure 14.** Time series of melt water discharge from GMM run November 2010 to November 2015 for the Fourcade Glacier, catchment of the Potter Cove, separated into the different source areas of snow, firn, ice and rock terrain ($q_{snow}$, $q_{firn}$, $q_{ice}$ and $q_{rock}$). The complete simulated melt water discharge ($q_{calc}$) is shown in black solid circle with the standard deviation of the time series.





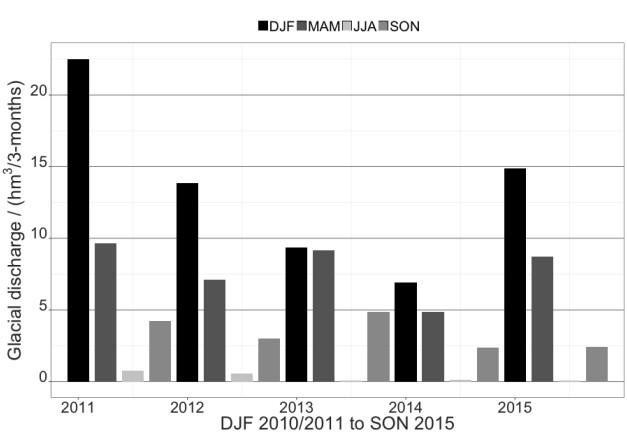

**Figure 15.** Seasonal melt water discharge from GMM run for the Fourcade Glacier, hydrological catchment of the Potter Cove (DJF: austral summer December - February; MAM: austral fall March - May; JJA: austral winter June - August; SON: austral spring September - November).



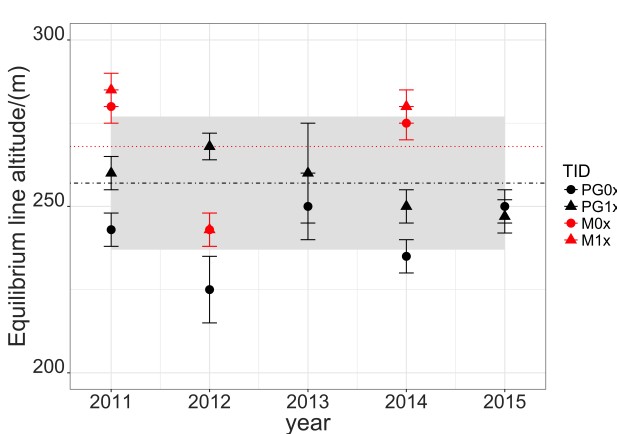

**Figure 16.** Equilibrium line altitude calculated from observations at transects PG0x (black, solid circle) and PG1x (black, solid triangle) and from GMM simulation (M) output at transects pixels PG0x (red, solid circle) and PG1x (red, solid triangle) for the time period 2010 to 2016.





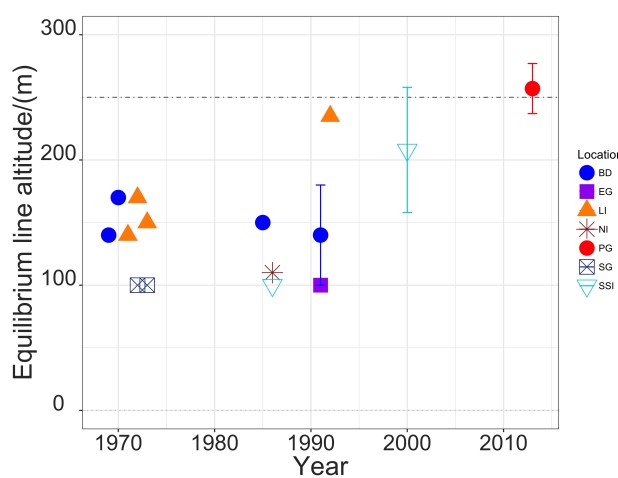

**Figure 17.** Equilibrium line altitude estimates from this study (PG) and former glaciological studies during the last decades. ELA estimates taken from literature encompass: BD: Bellingshausen Dome, KGI (Orheim and Govorukha (1982); Jiahong et al. (1994); Jiawen et al. (1995)), EG: Ecology Glacier, KGI (Bintanja, 1995), LI: Livingston Island (Vilaplana and Pallàs (1994); Molina et al. (2007); Navarro et al. (2009); Navarro et al. (2013)), NI: Nelson Island (Jiawen et al., 1995), SG: Stenhouse Glacier, KGI (Curl, 1980), SSI: South Shetland Islands (Serrano and López-Martínez, 2000)



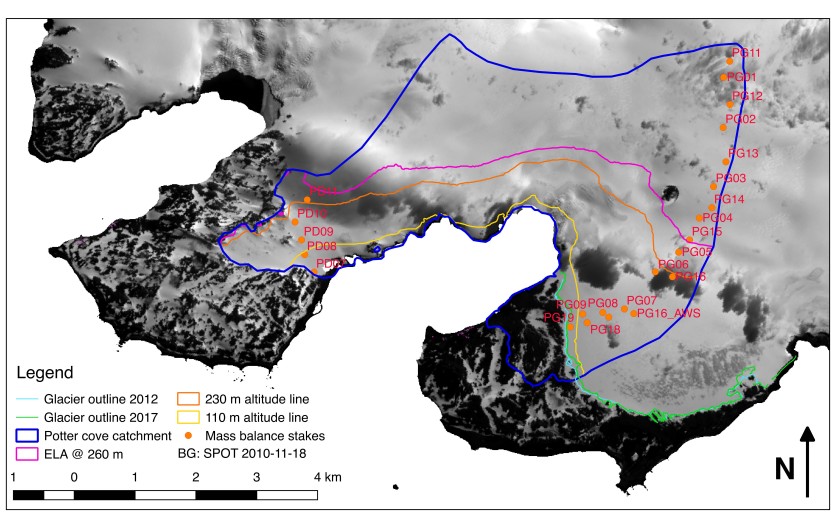

**Figure 18.** Glacier extent for equilibrium with actual climatic boundary conditions and an $ELA = 260$ m (pink line) for the Potter Cove catchment on KGI, according to an AAR between $AAR = 0.5$ (yellow line) and $AAR = 0.8$ (orange line). The actual glacier extent is marked as green and lightblue line and represents own Differential GPS measurements in austral summer 2012/2013 and 2016/17.



**Table 1.** Definition of beginning and end of the glaciological summer deducted from observations of local minima and maxima of the accumulation/ablation time series at the mass balance transects (PG0x and PG1x) on Potter Peninsula, King George Island, during the time period November 2010 to December 2016.

| Glaciological year | Begin of ablation period | | Begin of accumulation period | |
|---|---|---|---|---|
| | year | Julian day | year | Julian day |
| 2010/2011 | 2010 | 345 | 2011 | 103 |
| 2011/2012 | 2011 | 335 | 2012 | 85 |
| 2012/2013 | 2012 | 315 | 2013 | 139 |
| 2013/2014 | 2013 | 275 | 2014 | 74 |
| 2014/2015 | 2014 | 344 | 2015 | 126 |
| 2015/2016 | 2015 | 331 | 2016 | 110 |
| 2016/2017 | 2016 | 352 | | |