# Peer review of "Multi-year analysis of distributed glacier mass balance modelling and equilibrium line altitude on King George Island, Antarctic Peninsula"

_The Cryosphere, 2017_

## Referee Comment (RC1) · Anonymous Referee #1 · 18 Dec 2017

**Review of "Multi-year analysis of distributed glacier mass balance modelling and equilibrium line altitude on King George Island, Antarctic Peninsula", by Falk *et al*. (tc-2017-232)**

**Recommendation:** Requires major revision – not suitable for publication in current form.

**General**

This paper reports observation and model calculations of surface mass balance and meltwater production on a small glacier on King George Island, South Shetland Island. This glacier is situated in a region that has undergone rapid climate change in recent decades that has caused marked changes in the regional cryosphere. Cryospheric change provides useful information on climate change as it is an integrated response to a number of climatic variables. Furthermore, cryospheric change in this region has significant implications for terrestrial and marine ecosystems. The measurements reported in the paper have been made carefully and analysed in an appropriate fashion. Together with the model calculations of glacier mass balance and meltwater production, they provide a reliable indication of the "state of health" of Fourcade Glacier, and an indication of how its mass balance responds to short-term climate variability. These are useful and valuable results and are worthy of publication. However, I found the paper quite difficult to read because it is poorly structured. The motivation for the study is poorly presented, there is no clear statement of the scientific objectives (other than gathering data and carrying out a model study) and it is not clear what the major conclusions are. Furthermore, the authors have a tendency to include material on "methods" and "results" within the same section of the manuscript, which detracts from the clarity of the presentation. Below, I make some suggestions for restructuring and list some specific points that need attention to make the paper suitable for publication.

**Major (structural) points**

1. The scientific objectives of the study need to be clearly set out towards the end of the Introduction. At present there is just a list of the work that has been done. These objectives should be revisited in the Conclusions section.
2. Following section 2, "Study Area", I suggest a "Data and Methods" section structured as follows: (a) Meteorological measurements (AWS/Carlini/Bellingshausen), (b) Mass balance data, (c) Description of the GMM and its setup for this study. The long section on meteorological gap filling breaks the flow of this section – I would move it to an appendix or supplementary material. Note that the Data and Methods section should not include results (e.g. material on p11, lines 7-17) – such material belongs in the next section.
3. Next, a "Results" section presenting the main findings from the observations and model. As calibration of the model is reliant on the mass balance observations, I think it would be appropriate to discuss model calibration in this section.
4. Finally a Discussion/Conclusions section, framed in the context of the scientific objectives set out in the Introduction.

**Specific points**

P1 Abstract, l13: What do you mean by "no drift"?

P2, l1: Quantify "large fraction".

P2, l26: Sentence starting "The seasonal variability…" needs clarification. SAM needs to be defined.

P3, l12-16: Present as continuous text rather than a numbered list.

P3, l18: Give lat/lon for KGI. Refer to Fig 1 at this point (figure should ideally include a further map locating KGI with respect to the Antarctic Peninsula, South America, etc.).

P3 l30: Delete "in" before "especially".

P4, section 3.1.1: Mark AWS and temperature sensor locations on Fig. 1. Need to reference figures 2 and 3 (photographs) in this section.

P 4, l15: Insert "with" after "equipped".

P5, l6+: Discussion of the effects of pyroclastic debris does not belong in this section describing the measurements – move to the results section. Figure 4 also belongs in the results section, not here.

P7, l20: I don't understand why you applied a 48-hour smoothing to the cloud observations after interpolating to 1-hourly data. Surely you should use these data at the highest temporal resolution available (to make them compatible with your other driving data)?

P8, l9: Surely $m=Pa/(99*\cos(\psi))$ (assuming m is defined relative to 99 kPa) ?

P8, eqn. (9): Should also include a term for reflected longwave radiation = $(1-\varepsilon)*LW\downarrow$

P8, l25+: Give values for RMS differences between measured and modelled radiation components, as well as mean bias and r values. During winter Rn is typically in the range +/- 50 $Wm^{-2}$, the offset of 15.9 $Wm^{-2}$ apparent in eqn. (11) is really quite significant at this time of year.

P9, section 3.3: This section describes results so really belongs in section 4. (but needs to come before the section on model calibration). Include a table giving the elevations of the stakes.

P10, l30, and Fig. 8: What do the broken lines on Fig. 8 signify? The figure caption should state the reference date from which CMB has been calculated (i.e. the start date of the calculation, where CMB=0 for all stakes). You say that PG04 is in the accumulation zone, but there are hardly any measurements shown from this site and above – why not show results from at least one stake that is clearly within the accumulation zone? The overall trend at PG05 looks pretty close to zero, suggesting that this site is more or less on the equilibrium line.

P11, l10-17: Section 3.3 is concerned with surface mass balance observations. I think it is confusing to start talking about GMM results here before the GMM has been properly described.

P11, section 3.4: What is needed here is a section describing the GMM. Start by describing the model, then say how the model domain/catchment was set up and finish with a section on model validation against stake measurements.

P13, l17+: How can you be certain that the model error results form an underestimate of accumulation rather than an overestimate of ablation?

P13, l30: What do you mean by "a drift or disagreement … cannot be seen in the data"? Figs. 10 and 11 clearly show disagreement (hence drift) over the lower part of the catchment.

P14, eqn 14: Not sure why you show this equation – you are only able to measure the surface mass balance components.

P14, l24: "coverage". Why does high cloud coverage imply less precipitation and low ablation?

P16, l6: Briefly explain how you calculated ELA from observations and model.

P17, l24: I don't understand the sentence starting "If underneath the glacier is mountainous terrain…". The time taken for the glacier to disappear after the accumulation zone disappears depends on the magnitude of the ablation and the thickness of the glacier.

---

## Referee Comment (RC2) · Anonymous Referee #2 · 14 Jan 2018

This manuscript describes an interesting dataset of meteorological and glaciological data collected over the Patagonian Icefield over two glaciers, during 5 years, at short time intervals. The data is then used in order to process and calibrate a glacio-hydrological model that permits to assess the increase and amount of fresh melt water inputs into the sea/ocean due to a negative mass balance. This suggests strong impacts of climate change on the micro biota in the region but this is not described in detail. The scientific quality of the data and its relevance are both high, but unfortunately the structure of the text lacks organisation and shall be better put into perspective in order to smooth the reading and provide a more valuable input. I recommend this study to be published. The manuscript should undergo major revisions before that:

structure should be revised: results , methods, and data description are too much mixed resulting in a confusing overall text.

the introduction could be shortened a bit and resynthesized; how does the different changes in the climatic processes actually link to the melt/accumulation processes?

a broader perspective shall be introduced: for example:future predictions or discussion of the results in terms of future climate change...impact of these changes on the society/microbiota, etc...?

Specific comments

abstract

line 5 "distinct spatial heterogeneity reflecting the impact of synoptic[...]" I dont understand what you mean

line 6 "moist air with high temperatures and rain, and leads to melt conditions on the ice cap, fixating surface air temperatures to the melting point[...]" I believe you mean melting surface temperature? please rephrase

Introduction

line 1 "a large fraction" how much? page 3 line 16: add a point 6? projection?

Study area

the section could be synthesised

p3 line 32 "rarely absent" -> "frequent" ?  p3-4: "all year round" -> you said "rarely absent" just before so that is confusing p4 line 3: link that with the changes in the climatic systems evoked in the introduction

Datas and glaciological datasets

line 15 p4: are the radiation shield aspirated artificially? you should discuss if there's an impact of the radiation on the air temperature measurements.

p4 line 26: the nationality has no scientific significance

p6 line 9-11: might be moved to "methods"

p6 line 28: so what do you do in that case? could you explain?

p9 line 11-22: could you explain more precisely how you use your density measurement to interpret your snow height changes in terms of swe?

p10-11, until 3.4 : that mixes results/methods, please clarify.

p12 line 12: can you provide an estimate of the uncertainties on discharge outputs resulting from all the unknowns such as, for example, the bedrock topography? do you have any discharge measurements?

p12 line 30: how much that would impact further predictions (such as those made with the AAR)

p14 line 24: coverage

p15 line 2: to methods

p15 lines 6-10: to methods

line 21: ?

line 26 - onwards: to methods

p17: the AAR method should be described in a methods section

p18 line 19: See general comments: what would be the impact on micro biota of these changes (make links to the introduction)

---

## Author Comment (AC1) · 17 Jan 2018

**Response to Referee #1 comments:**
**Review of "Multi-year analysis of distributed glacier mass balance modelling and equilibrium line altitude on King George Island, Antarctic Peninsula", by Falk *et al*. (tc-2017-232)**

Response to major (structural) points:
We thank the reviewer for the thorough evaluation of our manuscript. The paper includes complex data sets and analysis, that has affeted the structure of the paper. We realize that it impacted the readability and agree to restructure the manuscript according to the referee's comments. In our experience, the meteorological data processing and gap filling is crucial for the results of the modelling work. Thus, we would like to keep it in the main text, but we agree that it can be condensed, shortened and relocated when necessary. Thanks again for the comprehensive reading and detailed advice.

Response to specific points:

P1 Abstract, l13: What do you mean by "no drift"?
We mean the deviation of model results from observations over the five-year period. The model is started with initial grid conditions, and it is a significant result that model and observations of glacier mass balance do not drift apart over time. We will rephrase this sentence.

P2, l1: Quantify "large fraction".
ca. 18% (17.995), we changed "large fraction" to "18%"

P2, l26: Sentence starting "The seasonal variability…" needs clarification. SAM needs to be defined.
The Southern Annular Mode (SAM) is a low-frequency mode of atmospheric variability that describes the north-south movement of the westerly wind belt around Antarctica. In recent years, the SAM has shown high positive numbers during autumn-winter which is associated with a contraction of the Antarctic high pressure cell and the circumpolar low pressure trough.  We added the definition to the manuscript.

P3, l12-16: Present as continuous text rather than a numbered list.
The numbered list is a different style that we have seen in other publications, but we don't object to the proposed change if this increases readability.

P3, l18: Give lat/lon for KGI. Refer to Fig 1 at this point (figure should ideally include a further map locating KGI with respect to the Antarctic Peninsula, South America, etc.).
Thanks for this advice. We can easily adapt the map to include the relative position of King George Island in the southern hemispherical context.

P3 l30: Delete "in" before "especially".
Done.

P4, section 3.1.1: Mark AWS and temperature sensor locations on Fig. 1. Need to reference figures 2 and 3 (photographs) in this section.
The AWS location is marked in the paper, although apparently not clearly enough. The additional temperature sensor locations are not marked since they were part of the former publication by Falk & Sala (2015), and air temperature lapse rates are discussed there in detail. We can add this information to the map.

P 4, l15: Insert "with" after "equipped".
Done.

P5, l6+: Discussion of the effects of pyroclastic debris does not belong in this section describing the measurements – move to the results section. Figure 4 also belongs in the results section, not here.
We put this remark here to explain the observations of low albedo. But we can see your point and will move this part to the results section.

P7, l20: I don't understand why you applied a 48-hour smoothing to the cloud observations after interpolating to 1-hourly data. Surely you should use these data at the highest temporal resolution available (to make them compatible with your other driving data)?
The linear interpolation between data points lead to a cloud cover curve that is very angular. This is not realistic and changes in cloud cover are more transitional. The smoothing was applied so that the resulting curve "looks" more realistic. It does not change the actual observational points much but the interpolated values. Cloud coverage at KGI in general is very high and the smoothing served only to shape the interpolated values to a less angular shape.

P8, l9: Surely $m = Pa/(99*cos(\psi))$ (assuming m is defined relative to 99 kPa)?
Yes, of course. This is a mistake, and we will correct it. Many thanks to the referee!

P8, eqn. (9): Should also include a term for reflected longwave radiation $= (1-\varepsilon)*LW\downarrow$
The term for longwave radiation flux towards the surface (downward) is represented by equation (8). Longwave radiation is absorbed by the cloud/atmosphere and then it is emitted as longwave radiation again from the atmosphere. This is not the same process as reflection. The total downward longwave radiation flux is calculated by considering the atmosphere as a black body radiator with a certain body temperature.

P8, l25+: Give values for RMS differences between measured and modelled radiation components, as well as mean bias and r values. During winter Rn is typically in the range +/- 50 Wm$_{-2}$, the offset of 15.9 Wm$_{-2}$ apparent in eqn. (11) is really quite significant at this time of year.
The referee is correct in this comment. RMSE are <=10 W/m2. Cloud coverage is typical for the area around the South Shetlands, even in winter. The meteorological data gathered at the Station shows an average of less than 10 clear sky days in most of the winters. This is reflected in the net radiation. We did not want to extend the meteorological data section more than necessary, but we agree that this is an important point. We will therefore add the statistics of differences and RMSE's.

P9, section 3.3: This section describes results so really belongs in section 4. (but needs to come before the section on model calibration). Include a table giving the elevations of the stakes.
This section comprises a detailed description of the glaciological observations and data time series. The last paragraph contains a comparison to the model output (p 11 ll 10-17), and indeed belongs to the results section. We included this paragraph here, since it is included in Fig. 8 to see the comparison of the model output for the stake locations to the observations. This is really interesting, but we agree that it belongs into the results section and will move the text part accordingly.

P10, l30, and Fig. 8: What do the broken lines on Fig. 8 signify? The figure caption should state the reference date from which CMB has been calculated (i.e. the start date of the calculation, where CMB=0 for all stakes). You say that PG04 is in the accumulation zone, but there are hardly any measurements shown from this site and above – why not show results from at least one stake that is clearly within the accumulation zone? The overall trend at PG05 looks pretty close to zero, suggesting that this site is more or less on the equilibrium line.
The broken lines represent an interpolation of the mass balance stake data points and should be

included in the legend. This is a mistake. We will update the figures accordingly.

P11, l10-17: Section 3.3 is concerned with surface mass balance observations. I think it is confusing to start talking about GMM results here before the GMM has been properly described.
The referee is right (as stated in the response above) and we will straighten this section.

P11, section 3.4: What is needed here is a section describing the GMM. Start by describing the model, then say how the model domain/catchment was set up and finish with a section on model validation against stake measurements.
The model description is included in section 3.5 (p 12 l 15 to p13 l 9), but the referee is right and the description should be BEFORE the description of model input and model calibration. We will move the model description to its own section as section 3.4, then input grids to the model as section 3.5, then calibration of the model as section 3.6. The current structure clearly reduces readability. Thanks a lot for making us aware of this flaw.

P13, l17+: How can you be certain that the model error results form an underestimate of accumulation rather than an overestimate of ablation?
The process that are not incorporated in the physical model are that of snow drift due to high wind speeds, and turbulence-driven snow deposition (also associated with high wind speeds). Refreezing processes are not included in the model physics since it was run in catchment configuration mode. These are all processes that are associated with accumulation of mass at the single grid points. Due to the high time resolution of the observations, we can clearly differentiate between different climatic periods, and thus feel confident enough to make this statement.

P13, l30: What do you mean by "a drift or disagreement … cannot be seen in the data"? Figs. 10 and 11 clearly show disagreement (hence drift) over the lower part of the catchment.
We discussed in the text the different processes leading to the difference between the model and the observations. Further, we discussed that the lowest stakes, PG09 and PG 19, are clearly subject to turbulence-driven snow deposition since located close to the glacier border and adjacent moraine. Periods of main differences are also associated with climatic conditions during late autumn and early winter. The overall behavior, though, shows no temporal accumulation of difference that would arise if model physics were not configured correctly. This can be properly seen in Fig. 8 and 9 that includes the spread of the GMM output for the stake locations to the MBS data time series.

P14, eqn 14: Not sure why you show this equation – you are only able to measure the surface mass balance components.
The goal was to put our results in the broader context of the mass balance, but we can adjust the manuscript in this point.

P14, l24: "coverage". Why does high cloud coverage imply less precipitation and low ablation?
The referee is right with this comment, and we did not intend this meaning. What we meant was that there was less precipitation (compared to other years) in form of rain leading to erosion of the snow and ice pack, therefore less ablation. High cloud coverage (meaning less global radiation) over the summer leads to less ablation due to less energy input to the surface.

P16, l6: Briefly explain how you calculated ELA from observations and model.
We interpolated the calculated net balance (bn) with a line, visually and by regression. The high variance of bn prevents an automated approach. The linear interpolation between data points close to the zero crossing is the most promising.

P17, l24: I don't understand the sentence starting "If underneath the glacier is mountainous terrain…". The time taken for the glacier to disappear after the accumulation zone disappears depends on the magnitude of the ablation and the thickness of the glacier.

We have to admit that our expression sounds a bit long-winded. Knowing the glacier surface elevation, the thickness of the glacier is determined by the bedrock. The magnitude of ablation depends on surface elevation. We mean that the time taken for the glacier to disappear after the accumulation zone disappears depends on the underlying bedrock, defining the thickness of the glacier and the magnitude of ablation.

---

## Author Comment (AC3) · 26 Jan 2018

**Response to Referee #2 comments:**
**Review of "Multi-year analysis of distributed glacier mass balance modelling and equilibrium line altitude on King George Island, Antarctic Peninsula", by Falk *et al*. (tc-2017-232)**

Response to major (structural) points:
We thank the reviewer for the careful evaluation of our manuscript. The paper includes complex data sets and analysis, that unfortunately has affected the structure of the paper. We acknowledge the comments and will straighten the manuscript to improve its structural integrity and readability.

To include future climate change scenarios as proposed would, in our opinion, result in a second paper that would deal with a sensitivity analysis of the modelling results. To our understanding, the presented research is already comprehensive and adding further elements would result in a too broad and diffuse paper.

Specific comments

**abstract**
line 5 "distinct spatial heterogeneity reflecting the impact of synoptic[...]" I dont understand what you mean
The rugged topography leads to a distinctly different extent of terrain exposure to the impact of the synoptic weather patterns. This then leads to a heterogeneous spatial pattern of ablation and accumulation areas of the glacier surface (more details in Falk et al. 2016).

line 6 "moist air with high temperatures and rain, and leads to melt conditions on the ice cap, fixating surface air temperatures to the melting point[...]" I believe you mean melting surface temperature? please rephrase
The melting point of a solid, in this case ice, is the temperature at which it changes state from solid to liquid at atmospheric pressure.

**Introduction**
line 1 "a large fraction" how much? page 3 line 16: add a point 6? projection?
The peripheral glaciers and icecaps of Antarctica represent ca. 18% of the global total. We changed the manuscript accordingly.

**Study area**
p3 line 32 "rarely absent" -> "frequent" ? p3-4: "all year round" -> you said "rarely absent" just before so that is confusing p4 line 3: link that with the changes in the climatic systems evoked in the introduction
"In general, days with temperatures above freezing are rarely absent in winter and are frequent in summer." This means, that air temperatures above freezing are present all year round, meaning also in winter, but more frequent of course during summer. This is recorded by the ice lenses in the snow pack of the accumulation zone.

**Datas and glaciological datasets**

line 15 p4: are the radiation shield aspirated artificially? you should discuss if there's an impact of the radiation on the air temperature measurements.

The radiation shields are not aspirated. This is a valid technique that assumes that wind speeds are high enough to ensure a natural ventilation. The location of the South Shetland in the Antarctic circumpolar low pressure through, as well as katabatic wind systems, leads to high wind speeds throughout the year. Ventilation is not an issue at the AWS due the high average wind speeds.

p4 line 26: the nationality has no scientific significance

The nationality of the scientific support is important for international collaboration and was added here for reasons of completeness.

p6 line 9-11: might be moved to "methods"

The reference to the glaciological model will be added to the model description section as part of the restructuring of the manuscript.

p6 line 28: so what do you do in that case? could you explain?

The battery voltage was logged with the meteorological data, so that we can differentiate between real features (like a frontal system) and fake sensor readings due to power failure. In the first case, we take the data as it is, of course. In the second case, we apply despiking routines.

p9 line 11-22: could you explain more precisely how you use your density measurement to interpret your snow height changes in terms of swe?

We are not sure if we understand the comment correctly. We use the snow density as a conversion factor to obtain the surface elevation changes in water equivalent.

p10-11, until 3.4: that mixes results/methods, please clarify.

In agreement with the comment of referee #1, the part p11 lines 9 to 17 will be moved to results.

p12 line 12: can you provide an estimate of the uncertainties on discharge outputs resulting from all the unknowns such as, for example, the bedrock topography? do you have any discharge measurements?

The discharge measurements were part of the hydrological research carried out by our Argentinean colleague, and is underway to being published. The values compare well for the time periods of hydrological observations, but these data cannot be used here in this manuscript. There is so far no information on bedrock topography. The basal melt is thus not included in the discharge that includes the spatially integrated melt and rain.

p12 line 30: how much that would impact further predictions (such as those made with the AAR)

Choosing higher values for the roughness length leads to higher calculated melt, and mostly to an overestimation of the calculated melt and discharge. We will add a short comment on this in the manuscript.

p14 line 24: coverage
This was a mistake and we corrected it. Thanks!

p15 line 2: to methods
We will move this part to the methods section as part of the restructuring of the manuscript. Thanks!

p15 lines 6-10: to methods
This is better located in the methods section, model description. Thanks for the comment!

line 21: ?
We are not quite sure, what the question mark refers to. p15 line 21: "The high variability of ablation and accumulation reflects the very high inter- and intra-annual variability of the meteorological boundary conditions (Falk and Sala, 2015)." This statement refers to the high inter- and intra-annual variability of accumulation and ablation observations as seen in Fig. 8 to 13. It is driven by the variability of the meteorological boundary conditions as discussed by Falk & Sala (2015).

line 26 - onwards: to methods
We do not agree with this comment. The correlation of the simulated discharge to positive degree days is clearly a result.

p17: the AAR method should be described in a methods section
The AAR is a standard index describing the health/status of a glacier. It is a simple definition as the ratio of accumulation area to total glacier area. We discuss here the results of analysis of our field data and modelling efforts by this index.

p18 line 19: See general comments: what would be the impact on micro biota of these changes (make links to the introduction)
We thank the author for this comment and will include a paragraph on the impact on biota to link with the introduction.

---

## Author Response (AR2)

**Response to Editor Decision: Publish subject to minor revisions**

We thank the editor and both referees for the throrough review. Their very useful comments certainly improved the quality and the readability of the manuscript.

**Minor revisions suggested by editor:**

First of all, we would like to let the editor know, that a paper investigating the governing processes, energy balance components and sensitivity analysis of the model in combination with eddy-covariance data is planned in the next step.

Specific remarks:

P13/L6: deleted „s" (typo)

P14/L1: deleted „physics"

P16/L4: now reads „... can be explained by changes..."

P18/L30: the editor is right with this comment and we changed „major impact" to „direct impact"

Re-read sections 5 and 6 carefully: Thanks a lot for this comment! We re-worked the two sections and moved a part of the conclusions to the discussion section. We therefore attach a latexdiff file, so that the editors can check the changes we did to the manuscript.

Figure 14: We added the sentence part „with the standard deviation of the time series as grey envelop" to the figure caption.

Figure 15: is now in colors

Figure 16: thanks for this comment. We re-did the graph so that all displayed features appear in the graph's legend and also changed the caption to improve readability of the graph.

Figure 18: this was a mistake in the legend entry and we fixed it, so that mass balance stakes are now included in the legend with symbol.

All figures using „CMB" in the axis titles: Thank you very much for this comment! We changed the figure captions accordingly. They now include the reference to the „cumulative mass balance (CMB)". We also changed the manuscript accordingly so now the abbreviation CMB is defined in the manuscripts' text.

Orders of figures: Thank you very much for this comment! During the work on the manuscripts, figure and section labels changed, which we tried to keep up with, but apparently we missed some labels. We checked all references to figures and sections and changed accordingly.

Thank you very much again for all the very useful comments during the review process. We are certain that it very much improved the quality and the readability of the manuscript!

best wishes,

Ulrike Falk, Damian Lopez, Adrian Silva-Busso

[revised manuscript text omitted]